# Relating Misfit to Gain in Weak-to-Strong Generalization Beyond the Squared Loss

**Abhijeet Mulgund** [* 1]   **Chirag Pabbaraju** [* 2]

## Abstract

The paradigm of weak-to-strong generalization constitutes the training of a strong AI model on data labeled by a weak AI model, with the goal that the strong model nevertheless outperforms its weak supervisor on the target task of interest. For the setting of real-valued regression with the squared loss, recent work quantitatively characterizes the gain in performance of the strong model over the weak model in terms of the misfit between the strong and weak model. We generalize such a characterization to learning tasks whose loss functions correspond to arbitrary Bregman divergences when the strong class is convex. This extends the misfit-based characterization of performance gain in weak-to-strong generalization to classification tasks, as the cross-entropy loss can be expressed in terms of a Bregman divergence. In most practical scenarios, however, the strong model class may not be convex. We therefore weaken this assumption and study weak-to-strong generalization for convex combinations of $k$ strong models in the strong class, in the concrete setting of classification. This allows us to obtain a similar misfit-based characterization of performance gain, up to an additional error term that vanishes as $k$ gets large. Our theoretical findings are supported by thorough experiments on synthetic as well as real-world datasets.

---

[*]Equal contribution [1]Department of Mathematics, Statistics, and Computer Science, University of Illinois at Chicago, Chicago, IL, USA [2]Department of Computer Science, Stanford University, Stanford, CA, USA. Correspondence to: Abhijeet Mulgund <mulgund2@uic.edu>, Chirag Pabbaraju <cpabbara@stanford.edu>.

*Proceedings of the $42^{nd}$ International Conference on Machine Learning*, Vancouver, Canada. PMLR 267, 2025. Copyright 2025 by the author(s).

Code is available at https://github.com/abhmul/general-misfit-gain.

## 1. Introduction

Weak-to-strong generalization (Burns et al., 2024) posits that strong AI models can outperform weaker supervisors when trained on data labeled by them. By virtue of being larger and more complex models that have presumably also seen more data in their pretraining lifecycle, the hope is that strong models will invariably know how to push past the performance ceiling of their weak supervisors. This phenomenon is central towards the eventual goal of superalignment (OpenAI, 2023), where we expect AI models to exhibit superhuman skills aligned with human principles from human supervision. Perhaps encouragingly, recent studies (Burns et al., 2024; Guo et al., 2024; Ji et al., 2024; Sun et al., 2024; Yang et al., 2024; Zhang et al., 2024; Liu & Alahi, 2024) have affirmed that such a phenomenon can in fact be elicited. Given the extent to which emergent AI would appear to influence our future going forward, it becomes imperative to theoretically understand when and how weak-to-strong generalization may provably be exhibited.

Towards this, Charikar et al. (2024) recently provided a mathematical characterization of weak-to-strong generalization for real-valued regression. They consider finetuning the final linear layer of a strong model (keeping other parameters fixed) to minimize the squared loss on labels given by a weak model. Using a geometric argument involving projections onto a convex set, Charikar et al. (2024) show that a strong model trained in this manner provably attains smaller loss than the weak model on the target task. Moreover, the quantitative reduction in loss (performance "gain") is at least as much as the loss of the strong model on the weakly labeled data ("misfit" between the strong and weak model). However, their results are limited to regression, leaving open whether such a misfit-based characterization can also be formally shown for other tasks like classification, one of the primary subjects of study in Burns et al. (2024).

Drawing on the theory of information geometry (Amari, 2016; Ay et al., 2017), we significantly extend the misfit-based characterization of weak-to-strong generalization beyond regression tasks. Our main observation is that the

---

This notion of misfit is also referred to as "student-supervisor disagreement" in Burns et al. (2024).

Pythagorean inequality for the squared loss used in Charikar et al. (2024) holds more generally for *Bregman divergences* (Bregman, 1967). Thus, whenever the loss function in the learning task is a Bregman divergence of some kind, a corresponding misfit-based weak-to-strong generalization inequality can be established. For example, since the squared $\ell_2$ distance is a Bregman divergence, the result of Charikar et al. (2024) constitutes a special case of this theory. More importantly, the standard loss function that is used in classification tasks, namely cross-entropy, can also be written down as a Bregman divergence (namely the Kullback-Leibler (KL) divergence (Kullback & Leibler, 1951)) plus an additive entropy term. This implies provable weak-to-strong generalization in the setting of classification (i.e. cross-entropy loss provably improves), along with a misfit-based characterization of the gain similar to Charikar et al. (2024).

Our approach, however, yields a non-standard recipe: over a convex class of strong models, minimize the expected KL divergence with the strong model's output as the *first* argument and weak labels as the second. This "reverse" KL is contrary to standard cross-entropy training, where the labels are the first argument. Our experiments, as well as concurrent work Yao et al. (2025a), suggest this reverse direction is better for weak-to-strong training for classification tasks. Furthermore, the reduction in loss is at least the KL divergence between the strong model and the weak model at the conclusion of weak-to-strong training! Thus, performance gain can yet again be measured in terms of the misfit between the strong and weak models, albeit with the notion of misfit being the KL divergence. Typically, however, the strong model class fails to be convex. In this case, we show that suitable approximation can be achieved by training a convex combination of $k$ strong models.

We validate our theory across synthetic as well as real-world NLP and vision datasets, and find that when the strong model is trained according to the stated recipe even with modest values of $k$ (e.g., 100), its performance gain is faithfully characterized by its KL misfit with the weak model. Additionally, this behavior becomes more apparent as $k$ increases, further validating our theory.

## 2. Related Work

Our work is complementary to a growing line of works, each of which seeks to theoretically explain the phenomenon of weak-to-strong generalization via a different lens. The work of Lang et al. (2024) posits that the two crucial properties governing weak-to-strong generalization are *coverage expansion* and *pseudolabel correction*. The work of Somerstep et al. (2024) formalizes weak-to-strong generalization as the problem of transferring a "latent concept" from the weak model to the strong model. Wu & Sahai (2024) show how finetuning a linear model on Gaussian features in the over-

parameterized regime provably exhibits weak-to-strong generalization. Shin et al. (2024) take a data-centric approach, and propose that data points that contain an overlap of *easy* as well as *hard* patterns most effectively elicit generalization. Ildiz et al. (2024) recently provide a high-dimensional analysis of knowledge distillation from surrogate models. Our work adopts the geometric viewpoint of Charikar et al. (2024), which interprets the finetuning of the strong model with weak labels as a *projection* of the weak model onto the strong model class (where the projection corresponds to minimizing a loss function).

Our work significantly generalizes Charikar et al. (2024) to more loss functions and non-convex strong model classes, providing stronger evidence that the gain in weak-to-strong generalization for many machine learning tasks is typically characterized by the disagreement between the student and teacher model (an observation originally made empirically by Burns et al. (2024)). This aligns well with the general flavor of results in co-training (Blum & Mitchell, 1998) and disagreement-based generalization bounds (Dasgupta et al., 2001; Wang & Zhou, 2017; Yu et al., 2019). Particularly relevant also is the vast literature on knowledge distillation (e.g., Hinton (2015); Nagarajan et al. (2023)) and self-distillation (e.g., Wei et al. (2020); Mobahi et al. (2020); Pareek et al. (2024)). It is worth mentioning that the work of Lang et al. (2024) draws insights from both Blum & Mitchell (1998) and Wei et al. (2020).

**Concurrent Work.** While we primarily focus on relaxing convexity assumptions in theorems relating misfit and gain, concurrent works Yao et al. (2025a;b) extend results relaxing the realizability assumption from Charikar et al. (2024) and establish a lower bound on gain by the misfit. It is worth noting that the results in Yao et al. (2025b) echo our observation that reverse KL divergence appears to be more suited for weak-to-strong generalization than forward KL divergence. Concurrent work Medvedev et al. (2025) studies the impact of early stopping on weak-to-strong generalization for regression in shallow ReLU networks.

## 3. Preliminaries

We begin by formally describing the weak-to-strong generalization setup, adopting notation from Charikar et al. (2024).

### 3.1. Weak-to-strong Generalization

The setup constitutes a "strong model" and a "weak model", where the strong model is typically a larger, and representationally more powerful AI model than the weak model. The weak model plays the role of teacher and the strong model plays the role of student, in that the strong model is trying to learn a target function from (potentially inaccurate) evaluations made by the weak model on data. Abstractly, we

think of the strong and weak models via their representation maps $h_s : \mathcal{X} \to \mathbb{R}^{d_s}$ and $h_w : \mathcal{X} \to \mathbb{R}^{d_w}$ respectively on the data domain $\mathcal{X}$. For example, $h_s$ could be a deep transformer architecture, and $h_w$ a shallow architecture. Suppose $g : \mathcal{X} \to \mathcal{Y}$ is some target function. The only signal that the strong model gets about $g$ is through evaluations of the weak model on data. Namely, it sees a dataset labeled by $f_w(h_w(\cdot))$, where $f_w : \mathbb{R}^{d_w} \to \mathcal{Y}$ is a finetuning map that the weak model has obtained, possibly after seeing a different batch of data labeled by $g$ itself. Importantly, the labels that the weak model feeds to the strong model is from a separately held-out dataset, so that the weak model does not have access to the true labels for it. The objective of the strong model then is to obtain a finetuning function $f_s$ for itself that it can compose onto its representation $h_s$, such that $f_s(h_s(\cdot))$ estimates $g$ *better* than $f_w(h_w(\cdot))$.

For the theory in the main body of the paper, we make two assumptions to avoid measure-theoretic and functional-analytic complications and to simplify the exposition: (1) we assume all distributions have finite support, and (2) we restrict our attention to binary classification rather than $c$-ary classification for $c > 2$. The main results can be generalized when (1) is relaxed; however, more involved technical machinery is required (see Appendix A.3). Only minor modifications are needed when (2) is relaxed (see Appendix A.2). We now set up some notation.

### 3.2. Notation

For $n \in \mathbb{N}$, we denote $[n] := \{1, \ldots, n\}$. $\overline{\mathbb{R}}^+ := \mathbb{R} \cup \{\infty\}$. For a function class $\mathcal{F}$ and a function $h$, we write $\mathcal{F} \circ h$ for the set $\{f \circ h\}_{f \in \mathcal{F}}$. For function $f$, we write $f \equiv c$ if $f$ is constantly $c$. For $p \in [0, 1]$, denote $\bar{p} := 1 - p$. All logarithms have base $e$. $H : [0, 1] \to [0, \log 2]$ is the binary Shannon entropy $H(p) := -p \log p - \bar{p} \log \bar{p}$. $D_{\text{KL}} : [0, 1]^2 \to \overline{\mathbb{R}}^+$ is the binary KL-divergence $D_{\text{KL}}(p\|q) := p \log \frac{p}{q} + \bar{p} \log \frac{\bar{p}}{\bar{q}}$, and $\text{XE} : [0, 1]^2 \to \overline{\mathbb{R}}^+$ is the binary cross-entropy $\text{XE}(p\|q) := -p \log q - \bar{p} \log \bar{q}$. Note $\text{XE}(p\|q) = D_{\text{KL}}(p\|q) + H(p)$. $\sigma : \mathbb{R} \to (0, 1)$ is the sigmoid function $\sigma(x) := \frac{e^x}{1+e^x}$, and $\sigma^{-1}$ is its inverse, the logit function.

For a subset $S \subset \mathbb{R}^n$, $\text{int}S$ denotes its interior, and $\overline{S}$ its closure. $\text{co}S$ is its convex hull which is the intersection of all convex sets containing $S$, and $\overline{\text{co}}S$ is its closed convex hull which is the intersection of all *closed* convex sets containing $S$. Note $\overline{\text{co}}S = \overline{\text{co}S}$. The convex hull of a set can be expressed as the set of all $k$-convex combinations of points in $S$, with $k$ ranging through $\mathbb{N}$. We use $\text{co}^k S$ to denote all convex combinations of any $k$ elements of $S$.

We use capital letters for random variables. These are instantiated by specifying a distribution (e.g., $X \sim P_X$ signifies random variable $X$ is drawn with probability distribution $P_X$). As mentioned earlier, all distributions are assumed

to have finite support. For a function $f$ of a random variable, $\mathbb{E}_X[f(X)]$ denotes the expectation of $f(X)$ over $P_X$. When no subscript is attached to $\mathbb{E}$, the expectation is taken with respect to all random variables in scope. The probability distribution of a random variable $X$ is written as $P_X$. For a pair of random variables $X, Y$ jointly distributed, the conditional distribution of $Y$ given $X$ is notated as $P_{Y|X}$.

### 3.3. Convexity

We will take for granted many basic results about convex functions; Ekeland & Temam (1999) is a good reference for these. For a convex function $\psi : \mathbb{R}^n \to \overline{\mathbb{R}}^+$, we write $\text{dom}\psi$ for $\{x \in \mathbb{R}^n : \psi(x) < \infty\}$. A convex function is *proper* if $\text{dom}\psi \neq \emptyset$. All convex functions in this paper will be assumed to be proper. We denote $U_\psi := \text{intdom}\psi$. Over $\mathbb{R}^n$, all convex functions with nonempty domain are continuous over $U_\psi$. When $\psi : \mathbb{R}^n \to \overline{\mathbb{R}}^+$ is strictly convex and $C^1(U_\psi)$, $\nabla\psi$ is a homeomorphism onto its image, and plays an important role in the theory of convex duality. When such a $\psi$ is specified, we denote $x^* := \nabla\psi(x)$ for $x \in U_\psi$. Likewise, for $S \subset U_\psi$, $S^* := \nabla\psi(S)$. The Legendre dual of such a $\psi$ is $\psi^* : \mathbb{R}^n \to \overline{\mathbb{R}}^+$, where $\psi^*(x^*) := \langle x, x^* \rangle - \psi(x)$ for $x \in U_\psi$, and $\infty$ otherwise. It is also a strictly convex function that is $C^1$ on $U_{\psi^*} = (U_\psi)^*$. Furthermore, $\psi^{**} = \psi$, and $\nabla\psi^* = (\nabla\psi)^{-1}$. As such, When $\psi$ and $\psi^*$ are distinguished, we call $U_\psi$ the primal space and $U_{\psi^*}$ the dual space. We refer to $\nabla\psi$ as the dual map, and $x^*$ the dual of $x \in U_\psi$.

### 3.4. Bregman Divergences

We generalize beyond the squared loss analyzed in Charikar et al. (2024) via Bregman divergences (Bregman, 1967).

**Definition 3.1** (Bregman Divergence). Let $\psi : \mathbb{R}^n \to \overline{\mathbb{R}}^+$ be a strictly convex and $C^1(U_\psi)$. Then the $\psi$-Bregman divergence, $D_\psi : \mathbb{R}^n \times U_\psi \to \overline{\mathbb{R}}^+$ is defined

$$D_\psi(x, y) := \psi(x) - \psi(y) - \langle x - y, y^* \rangle$$
$$= \psi(x) + \psi^*(y^*) - \langle x, y^* \rangle.$$

Here, $\psi$ is the *generator* of $D_\psi$. Intuitively, $D_\psi$ measures how much the linear approximation of $\psi$ at $y$ underestimates $\psi(x)$. It is always nonnegative, and is 0 if and only if $x = y$. The alternative expression for $D_\psi$ also reveals an important property: $D_\psi(x, y) = D_{\psi^*}(y^*, x^*)$. Furthermore, $D_\psi$ is strictly convex and differentiable in its first argument.

Table 2 in Appendix A.1 lists some examples of Bregman divergences. While Bregman divergences are convex in their first argument, they are *not necessarily* convex in their second argument. The logistic cost (Table 2) is an example of this.

### 3.4.1. CROSS-ENTROPY AND KL-DIVERGENCE

In binary classification tasks, we work with data $X, Y \sim P_{X,Y}$, where input data $X \in \mathbb{R}^d$ and labels $Y \in \{0, 1\}$. Let $g : \mathbb{R}^d \to [0, 1]$ be the conditional probability function $g(x) := P_{Y|X}(1|x)$. Given some hypothesis class $\mathcal{F}$ of functions $\mathbb{R}^d \to (0, 1)$, we wish to find an $f \in \mathcal{F}$ that minimizes $\mathbb{E}[\mathrm{XE}(g(X)\|f(X))]$. Since $g$ is fixed for a given classification task, we see that $\mathbb{E}[\mathrm{XE}(g(X)\|f(X))]$ differs from $\mathbb{E}[D_{\mathrm{KL}}(g(X)\|f(X))]$ by the task-dependent constant $\mathbb{E}[H(g(X))]$. Thus, for classification tasks, *applying Bregman divergence theory will produce corresponding results about cross-entropy, modulo a constant*.

### 3.4.2. GEOMETRY OF BREGMAN DIVERGENCES

We would like to use Bregman divergences analogously to distances like the $\ell_2$ distance. However, Bregman divergences are not always symmetric. The KL-divergence is a classic example of a non-symmetric Bregman divergence. Thus, they are not valid distances. However, they do possess many geometric properties. We use two properties below to generalize the main result from Charikar et al. (2024). Below, $\psi$ refers to a strictly convex, $C^1(U_\psi)$ function.

**Fact 3.2** (Generalized Law of Cosines (Chen & Teboulle, 1993)). *Let $x, y, z \in U_\psi$. Then*

$$D_\psi(x, z) = D_\psi(x, y) + D_\psi(y, z) - \langle x - y, z^* - y^* \rangle.$$

To generalize the Pythagorean inequality for inner product spaces, we need the following notion

**Definition 3.3.** Let $\mathcal{W} \subset U_\psi$. For $x \in U_\psi$, define the forward Bregman projection to be $P_\mathcal{W}(x) := \arg\min_{y \in \mathcal{W}} D_\psi(y, x)$.

If $\mathcal{W}$ is convex and closed then the forward projection exists and is unique. Existence and uniqueness follow from continuity, boundedness of sublevel sets (Exercise 1 in (Fawzi, 2022)), and strict convexity of $D_\psi$ in its first argument. The following is a known generalization of the standard $\ell_2$ Pythagorean inequality to Bregman divergences.

**Fact 3.4** (Generalized Pythagorean Inequality (Dhillon, 2007)). *Let $\mathcal{W} \subset U_\psi$ be a closed, convex set. Then for all $z \in U_\psi$ and $x \in \mathcal{W}$*

$$D_\psi(x, z) \geq D_\psi(x, P_\mathcal{W}(z)) + D_\psi(P_\mathcal{W}(z), z)$$

### 3.4.3. EXPECTATIONS OF BREGMAN DIVERGENCES

Suppose we have a random variable $X \sim P_X$ with finite support $\mathcal{X}$. Functions $\mathcal{X} \to \mathbb{R}^n$ form an $n|\mathcal{X}|$ dimensional vector space that we give the $L^2$ norm $\sqrt{\mathbb{E}\left[\|f(X) - g(X)\|_2^2\right]}$. From strictly convex, $C^1$, $\psi : \mathbb{R}^n \to \overline{\mathbb{R}}^+$, we get a convex functional $\Psi$ defined $\Psi[f] := \mathbb{E}[\psi(f(X))]$. Computing

the dual map, Legendre dual, and Bregman divergence of $\Psi$, we get $f^* = \nabla\psi \circ f$, $\Psi^*[f^*] = \mathbb{E}\psi^*(f^*(X))$, and $D_\Psi(f, g) = \mathbb{E}D_\psi(f(X), g(X))$. Thus, the previous results about Bregman divergences also apply to *expectations* of Bregman divergences when $|\mathcal{X}| < \infty$. See Appendix Appendix A.3 for a discussion when $\mathcal{X}$ is not finite.

### 3.4.4. BREGMAN DIVERGENCES IN THE WILD

A long line of literature has established the value of Bregman divergences in machine learning. Perhaps the most well-known application is to the mirror-descent algorithm, which generalizes the gradient descent algorithm to non-Euclidean geometries Gupta (2020). This generalization can lead to improved bounds on regret for online learning tasks, and unifies several optimization algorithms under one framework. Similarly, in unsupervised learning, Banerjee et al. (2005) unifies a wide variety of clustering algorithms as arising from a particular choices of Bregman divergences. Furthermore, they show every exponential family for soft-clusering corresponds to an expectation-maximization algorithm using a Bregman divergence. In statistics, Reid & Williamson (2011) establish a connection between Bregman divergences and $f$-divergences.

In this paper, we are primarily focused on establishing a misfit-gain inequality (Corollary 4.2) for cross-entropy, and generalizing it to non-convex classes of strong learners (Theorem 4.3). However, we frame these theorems and their proofs (Appendices A.2.1 and A.2.2) through the lens of Bregman divergence theory to provide a toolbox for researchers looking to apply these results to more specialized settings.

### 3.4.5. FORWARD KL VS. REVERSE KL

Here we provide a brief intuitive explanation of the difference between forward and reverse KL divergence. We echo observations made in Kristiadi (2016); Yao et al. (2025a); readers should refer to these sources for more detailed discussion.

Roughly, we can think of forward KL as being *mass-seeking*: it prioritizes learning a distribution that covers all possibilities dictated by the teacher. On the other hand, reverse KL is *mode-seeking*: it prioritizes learning a distribution that captures the most frequent behavior in the teacher. One can notice this behavior in how the two loss functions handle a student that disagrees with the teacher and predicts 0% probability for an event: forward KL will be $+\infty$ while reverse KL will exclude that event from the loss. Conversely if the teacher predicts 0% probability for an event and the student disagrees, then forward KL will disregard that event, reverse KL will become $+\infty$.

In the context of weak-to-strong generalization, it is plausi-

ble that the (weak) teacher makes errors. Thus, we do not want our (strong) student to be mass-seeking: it should be free to disagree with its teacher. Instead, the student should be *mode-seeking* as this is likely where most of the signal from the teacher comes from.

# 4. Main Results

We are now in a position to state our main results on weak-to-strong generalization using Bregman divergence theory.

The first result is a direct application of Fact 3.4.

**Theorem 4.1** (Bregman Misfit-Gain Inequality). *Let $\psi : \mathbb{R}^n \to \overline{\mathbb{R}}^+$ be a proper convex function s.t. $U_\psi \neq \emptyset$. Let $h_s : \mathcal{X} \to \mathbb{R}^{d_s}$ and $h_w : \mathcal{X} \to \mathbb{R}^{d_w}$ be the strong and weak learner representations respectively. Let $f_w : \mathbb{R}^{d_w} \to U_\psi$ be the weak model finetune layer, and $g : \mathcal{X} \to U_\psi$ be the target function. Let $\mathcal{F}$ be a class of functions mapping $\mathbb{R}^{d_s} \to U_\psi$. If the following hold:*

1. *(Realizability) $\exists f_* \in \mathcal{F}$ s.t. $g = f_* \circ h_s$,*

2. *(Convexity) $\mathcal{F}$ is a convex set of functions,*

3. *(Sequential Consistency) For $y \in U_\psi$ fixed, if $D_\psi(x_n, y) \to 0$, then $x_n \to y$,*

*then for any $\epsilon > 0$, there exists $\delta > 0$ such that for all $f_s \in \mathcal{F}$ that satisfy*

$$\mathbb{E}\big[D_\psi\left(f_s(h_s(X)), f_w(h_w(X))\right)\big] \\ \leq \inf_{f \in \mathcal{F}} \mathbb{E}\left[D_\psi\left(f(h_s(X)), f_w(h_w(X))\right)\right] + \delta,$$

*we have*

$$\mathbb{E}\big[D_\psi\left(g(X), f_s(h_s(X))\right)\big] \\ \leq \mathbb{E}\left[D_\psi\left(g(X), f_w(h_w(X))\right)\right] \\ - \mathbb{E}\left[D_\psi\left(f_s(h_s(X)), f_w(h_w(X))\right)\right] + \epsilon. \quad (1)$$

*Proof Sketch.* Because $\mathcal{F}$ is convex, we get that $\mathcal{F} \circ h_s$ is also convex. Since $g \in \overline{\mathcal{F} \circ h_s}$, by Fact 3.4, we have that $g_{\text{proj}} := P_{\overline{\mathcal{F} \circ h_s}}(f_w \circ h_w)$ uniquely exists and satisfies the Bregman Pythagorean inequality. Now, by continuity of both $\mathbb{E}[D_\psi(\cdot, f_w(h_w(X)))]$ and $\mathbb{E}[D_\psi(g(X), \cdot)]$ there exists an $\epsilon_1 > 0$ s.t. if $\mathbb{E}\left[\|f(h_s(X)) - g_{\text{proj}}(X)\|_2^2\right] < \epsilon_1$, then $f$ almost satisfies the Pythagorean inequality with error $\epsilon$. Choosing $\delta$ sufficiently small, we can apply the Pythagorean inequality to bound $\mathbb{E}[D_\psi(f(h_s(X)), g_{\text{proj}}(X))]$. Sequential consistency will then make $\mathbb{E}\left[\|f(h_s(X)) - g_{\text{proj}}(X)\|_2^2\right] < \epsilon_1$. $\quad \square$

Sequential consistency is a common assumption in the literature when defining Bregman divergences (Reem et al., 2018; Bauschke & Combettes, 2003; Butnariu et al., 2003), and is satisfied with very weak assumptions on $\psi$. Theorem 4.1 generalizes Theorem 1 in Charikar et al. (2024), since $\ell_2$ is a Bregman divergence (and clearly sequentially consistent).

As a corollary to Theorem 4.1, we also obtain the misfit-gain inequality for cross-entropy.

**Corollary 4.2** (Cross-Entropy Misfit-Gain Inequality). *Let $h_s : \mathcal{X} \to \mathbb{R}^{d_s}$ and $h_w : \mathcal{X} \to \mathbb{R}^{d_w}$ be the strong and weak model representations respectively. Let $g : \mathcal{X} \to (0, 1)$ be the target function. Let $\mathcal{F}$ be a class of functions mapping $\mathbb{R}^{d_s} \to (0, 1)$. Let $f_w : \mathbb{R}^{d_w} \to (0, 1)$ be the classifier for the weak model. If the following hold:*

1. *(Realizability) $\exists f_* \in \mathcal{F}$ so that $g = f_* \circ h_s$,*

2. *(Convexity) $\mathcal{F}$ is a convex set of functions,*

*then for any $\epsilon > 0$, there exists $\delta > 0$ such that for all $f_s \in \mathcal{F}$ that satisfy*

$$\mathbb{E}\big[D_{\text{KL}}(f_s(h_s(X))\|f_w(h_w(X)))\big] \\ \leq \inf_{f \in \mathcal{F}} \mathbb{E}\left[D_{\text{KL}}\left(f(h_s(X))\|f_w(h_w(X))\right)\right] + \delta,$$

*we have*

$$\mathbb{E}\big[\text{XE}(g(X)\|f_s(h_s(X)))\big] \leq \mathbb{E}\left[\text{XE}\left(g(X)\|f_w(h_w(X))\right)\right] \\ - \mathbb{E}\left[D_{\text{KL}}\left(f_s(h_s(X))\|f_w(h_w(X))\right)\right] + \epsilon. \quad (2)$$

*Proof Sketch.* We can first rewrite Equation (2) in terms of $D_{\text{KL}}$ instead of XE by subtracting $H(g(X))$ from both sides. Since $D_{\text{KL}}$ is sequentially consistent by Pinsker's inequality, we can apply Theorem 4.1. $\quad \square$

We note that if

$$f_s = \arg\min_{f \in \mathcal{F}} \mathbb{E}\left[D_{\text{KL}}(f(h_s(X))\|f_w(h_w(X)))\right],$$

then Equation (2) directly holds with no $\epsilon$ term. Such an $f_s$ uniquely exists if $\mathcal{F} \circ h_s$ is closed. We refer to $\mathbb{E}\left[D_{\text{KL}}(f_s(h_s(X))\|f_w(h_w(X)))\right]$ as the "misfit" term and $\mathbb{E}\left[\text{XE}(g(X)\|f_w(h_w(X)))\right] - \mathbb{E}\left[\text{XE}(g(X)\|f_s(h_s(X)))\right]$ as the "gain" term.

In practice, $\mathcal{F}$ is usually not convex. For example, when learning a linear probe (Burns et al., 2024) on top of $h_s$ together with the sigmoid, $\mathcal{F}^*$ is convex, but $\mathcal{F}$ itself is not. Nevertheless, our main observation in this case is that Theorem 4.2 still applies to $\text{co}\mathcal{F}$! By Caratheodory's theorem, $\text{co}\mathcal{F} = \text{co}^{|\mathcal{X}|+1}\mathcal{F}$, and hence we can simply consider convex combinations of $|\mathcal{X}| + 1$ functions in $\mathcal{F}$ to represent $\text{co}\mathcal{F}$. However, as $|\mathcal{X}| \to \infty$, this representation is not computationally tractable. We therefore suggest remedying this by attempting to project the weak model onto $\text{co}^k \mathcal{F} \circ h_s$, and show that the error in the misfit-gain inequality can still be bounded by a decreasing function in $k$ that is independent of $|\mathcal{X}|$. Concretely, we show that:

**Theorem 4.3.** *Let $h_s, h_w, f_w, g$ be as in Theorem 4.2. Let $\mathcal{F}$ be a class of functions mapping $\mathbb{R}^{d_s} \to (0,1)$. If the following hold*

1. *(Realizability) $\exists f_* \in \mathcal{F}$ so that $g = f_* \circ h_s$,*

2. *(Regularization) $\mathcal{F}$ satisfies*

$$\max\left(\mathbb{E}\left[\sup_{f \in \mathcal{F}} 1/f(X)\right], \mathbb{E}\left[\sup_{f \in \mathcal{F}} 1/\overline{f(X)}\right]\right) < \infty,$$

*then for any $k \in \mathbb{N}$, there exists $\delta > 0$ such that for all $f_s \in \mathrm{co}^k \mathcal{F}$ that satisfy*

$$\mathbb{E}\big[D_{\mathrm{KL}}(f_s(h_s(X))\|f_w(h_w(X)))\big]$$
$$\leq \inf_{f \in \mathrm{co}^k \mathcal{F}} \mathbb{E}\left[D_{\mathrm{KL}}(f(h_s(X))\|f_w(h_w(X)))\right] + \delta,$$

*we have*

$$\mathbb{E}\big[\mathrm{XE}(g(X)\|f_s(h_s(X)))\big] \leq \mathbb{E}\left[\mathrm{XE}(g(X)\|f_w(h_w(X)))\right]$$
$$- \mathbb{E}\left[D_{\mathrm{KL}}(f_s(h_s(X))\|f_w(h_w(X)))\right] + O(1/\sqrt{k}). \quad (3)$$

The proof of Theorem 4.3, inspired from the work of Zeevi & Meir (1997), is more involved, and is a careful application of the probabilistic method; we defer it to the Appendix as Theorem A.2 where we prove it in the multi-class setting; in particular, with $c$ classes, the error term becomes $O(\sqrt{c/k})$.

We note that both Corollary 4.2 and Theorem 4.3 make no assumption on the weak model, and only the realizability assumption on the target. Theorem 4.3 makes only a mild assumption on $\mathcal{F}$ that can be enforced by regularizing the models in $\mathcal{F}$ (e.g., via an $\ell_2$ penalty). We emphasize however that the inequality does not guarantee *significant* weak-to-strong generalization for any weak model. If $f_w \in \mathcal{F}$, then the $f_s$ obtained by either theorems will be $f_w$ itself, and the misfit term becomes 0. However, Corollary 4.2 and Theorem 4.3 allow us to *quantify* how much weak-to-strong generalization we should expect to see. All expectations involved can be estimated on a hold-out dataset, yielding a bound on the gain in terms of empirical misfit, modulo estimation error. This quantification is in the *loss* of the learned strong model relative to the (realizable) ground truth data. Since a reasonable loss function is correlated with other error metrics like accuracy, we should expect to empirically see improvements in accuracy with increases in loss misfit. However, a relationship between accuracy and loss misfit is *not* theoretically guaranteed.

Theorem 4.3 provides a concrete recipe for weak-to-strong generalization that allows for a quantitative handle on the performance gap between the weak and strong model. In this recipe, we finetune a convex combination of $k$ functions from $\mathcal{F}$ on top of the strong model representation. The objective of this finetuning is the empirical mean of the KL divergence between the strong and weak model output. Importantly, the strong model's output is in the *first* argument of the KL divergence. This is in contradistinction to the standard weak supervision objective, wherein the weak supervisor's output is in the first argument. However, this distinction provably leads to a performance gain, provided $k$ is not too small, and the empirical estimates are accurate. In the next section, we empirically validate this recipe.

## 5. Experiments

We conduct experiments on synthetic data similar to Charikar et al. (2024), and also NLP and vision datasets considered originally in the work of Burns et al. (2024).

### 5.1. Synthetic Data Experiments

We follow the setup in Charikar et al. (2024) for the synthetic experiments. We assume that the target $g$ takes the form $f(h^\star(\cdot))$ for some ground-truth representation map $h^\star$ and finetuning function $f \in \mathcal{F}$. We set $h^\star : \mathbb{R}^8 \to \mathbb{R}^{16}$ to a randomly initialized MLP with 5 hidden layers and ReLU activations, where the hidden size is 16. We choose $\mathcal{F}$ to be the set of $c$-class logistic regression models on $\mathbb{R}^{16}$. Namely,

$$\mathcal{F} = \{x \mapsto \sigma(Wx) : W \in \mathbb{R}^{(c-1)\times 16}\},$$
where $\sigma(z_1^*, \ldots, z_{c-1}^*) := \mathrm{softmax}([0, z_1^*, \ldots, z_{c-1}^*]).$

The marginal $\mathcal{P}$ of the data is $\mathcal{N}(0, \nu^2 I)$; we set $\nu = 100$.

**Pretraining.** The class of strong and weak model representations, $\mathcal{H}_s$ and $\mathcal{H}_w$, are respectively set to the class of 8-layer and 2-layer MLPs mapping $\mathbb{R}^8 \to \mathbb{R}^{16}$, again with ReLU activations and hidden layer size 16. To obtain $h_s$ and $h_w$, we first randomly sample $T = 10$ maps $f^{(1)}, \ldots, f^{(T)}$ from $\mathcal{F}$, and generate data $\{x_j^{(i)}, y_j^{(i)}\}_{j=1}^{N_r}$ for each, where $N_r = 2000$. Here, every $x_j^{(i)} \sim \mathcal{P}$, and $y_j^{(i)} = f^{(i)}(h^\star(x_j^{(i)}))$. Thereafter, the parameters of $h_s$ and $h_w$ are obtained by performing gradient descent to optimize the cross-entropy loss:

$$h_k = \arg\min_{h \in \mathcal{H}_k} \frac{1}{TN_r} \sum_{i=1}^{T} \sum_{j=1}^{N_r} \mathrm{XE}(y_j^{(i)}\|f^{(i)}(h(x_j^{(i)})))$$
$$\text{for } k \in \{w, s\}. \quad (4)$$

**Weak Model Finetuning.** Next, we finetune the weak model on $M = 100$ fresh finetuning tasks $f^{(1)}, \ldots, f^{(M)}$ from $\mathcal{F}$. To do so, we again generate $\{x_j^{(i)}, y_j^{(i)}\}_{j=1}^{N_f}$ for every $i \in [M]$, where $N_f = 2000$, each $x_j^{(i)} \sim \mathcal{P}$ and $y_j^{(i)} = f^{(i)}(h^\star(x_j^{(i)}))$. The weak model representation $h_w$ that was obtained in the pretraining step is held frozen, and the parameters in the final linear layer are obtained via

gradient descent to minimize the cross-entropy loss:

$$f_w^{(i)} = \arg\min_{f \in \mathcal{F}} \frac{1}{N_f} \sum_{j=1}^{N_f} \mathrm{XE}(y_j^{(i)} \| f(h_w(x_j^{(i)}))). \quad (5)$$

**Weak-to-Strong Supervision.** For each finetuning task above, we obtain data labeled by the weak supervisor as follows. For every $i \in [M]$, we generate $\{\tilde{x}_j^{(i)}, \tilde{y}_j^{(i)}\}_{j=1}^{N_f}$, where $\tilde{x}_j^{(i)} \sim \mathcal{P}$ and $\tilde{y}_j^{(i)} = f_w^{(i)}(h_w(\tilde{x}_j^{(i)}))$. This data is fed to the strong model, which goes on to learn:

$$f_s^{(i)} = \arg\min_{\substack{f_1,\ldots,f_k \in \mathcal{F} \\ \lambda \in (0,1)^k : \sum_a \lambda_a = 1}} \frac{1}{N_f} \sum_{j=1}^{N_f} D_{\mathrm{KL}} \left( \sum_{a=1}^{k} \lambda_a z_a^{(i,j)} \,\Big\|\, \tilde{y}_j^{(i)} \right),$$

$$\text{where} \quad z_a^{(i,j)} = f_a(h_s(\tilde{x}_j^{(i)})). \quad (6)$$

Namely, we optimize over a convex combination of $k$ logistic regression heads. Importantly, observe that the order of arguments in $D_{\mathrm{KL}}$ in Equation (6) above is flipped from that in Equation (5), in keeping with the recipe given by Theorem 4.3. We set $k = 100$.

**Evaluation.** To evaluate that the inequality given by Theorem 4.3 holds, for each task $i \in [M]$, we estimate the three expectations in Equation (3) from a new sample of size $N_f$. Our result says that upto an error term of $O(\sqrt{c/k})$,

$$\mathbb{E}\left[\mathrm{XE}(g(X)\|f_w(h_w(X)))\right] - \mathbb{E}\left[\mathrm{XE}(g(X)\|f_s(h_s(X)))\right]$$
$$\geq \mathbb{E}\left[D_{\mathrm{KL}}(f_s(h_s(X))\|f_w(h_w(X)))\right]. \quad (7)$$

In Figure 1, we plot the LHS on the y-axis and the RHS on the x-axis, for the experiment above performed with $c = 2, 10, 50, 100$. We can observe that Equation (7) is exhibited more or less with equality, for both the binary ($c = 2$) as well as multiclass cases ($c > 2$). We do note that the plot gets noisier as $c$ grows to 100. These plots show the same trend as for the squared loss in Charikar et al. (2024).

### 5.2. NLP Tasks

Next, we consider four real-world NLP classification datasets: BoolQ (Clark et al., 2019), SciQ (Welbl et al., 2017), CosmosQA (Huang et al., 2019) and Amazon Polarity (McAuley & Leskovec, 2013). We work with models in the gpt2 family (Radford et al., 2019), where the weak model is fixed to be gpt2, and the strong model is chosen

---

These are four of the five datasets considered in the codebase provided by OpenAI (Ecoffet et al., 2023) for their weak-to-strong generalization paper. We chose to not include the fifth (Anthropic/HH-RLHF (Bai et al., 2022; Ganguli et al., 2022)) since the results of Burns et al. (2024) on this dataset are extremely noisy (see the plot in Ecoffet et al. (2023)), e.g., no model seems to be doing better than random guessing.

from gpt2-medium, gpt2-large and gpt2-xl. For each dataset, we first finetune the linear probe of the weak model (by minimizing cross-entropy) on 50% of the training data with ground-truth labels. We then compute weak labels given by the trained model on the remaining 50% of the training data. We then optimize a convex combination of $k = 100$ logistic regression heads on top of the strong model to minimize the reverse objective as in Equation (6). We add an $\ell_2$ regularization penalty (with coefficient 0.1) on the linear weights in the objective to help with regularization, and to better align with the requirements of Theorem 4.3. Finally, we estimate each of the terms in Equation (7) from the test data. The obtained results are shown in Figure 2.

Firstly, we see weak-to-strong generalization (in terms of the loss) for all the datasets: the test loss of the strong model is always smaller than the test loss of its weak supervisor. Next, we see a clear trend on all four datasets: as we range the strong model from gpt2-medium to gpt2-large to gpt2-xl, the misfit onto the weak model increases, and concurrently, the loss on the test data decreases. In fact, for CosmosQA, Amazon Polarity and SciQ, in addition to the test loss decreasing, we observe that the test *accuracy* is non-decreasing too (note again that our result does not claim anything about accuracy per se).

We remark that our accuracies are inferior to those reported in the experiments by Burns et al. (2024) on these datasets. One reason for this is that we only ever tune the logistic regression heads of models (in both weak model training, as well as weak-to-strong training), whereas Burns et al. (2024) allow full finetuning that includes the representation layer parameters.

### 5.3. Vision Tasks

We next perform experiments on image classification datasets. Following Burns et al. (2024), we fix the weak supervisor to be AlexNet (Krizhevsky et al., 2012). For the strong model, we consider ResNet-50 (He et al., 2016) and ViT-B/8 (Dosovitskiy, 2020) based on DINO representations (Caron et al., 2021). Having fixed these representations, we finetune a convex combination of $k = 100$ logistic heads on top of the strong model on the weakly labeled data. We consider two datasets: CIFAR-10 (Krizhevsky et al., 2009) and ImageNet (Russakovsky et al., 2015), and obtain the same plots as we did for the NLP datasets in Figures 3a, 3b. Again, we clearly observe that as the misfit of the strong model onto the weak labels increases, both, the test loss decreases as well as the test accuracy increases! Interestingly, we observe that our weak-to-strong accuracy on ImageNet for ViT-B/8 is *better than* the corresponding weak-to-strong accuracy reported for the same experiment in Table 3, Burns et al. (2024) (namely 64.2% respectively). We note that the only difference in our setup is the weak-to-strong training

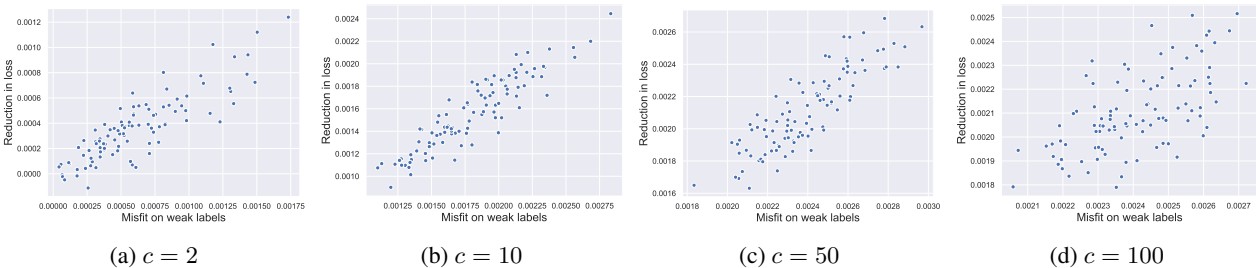

Figure 1. Synthetic data experiments. The Gain and Misfit closely track each other. For $c = 100$, we see that the correlation between misfit and gain weakens, suggested also by the $O(\sqrt{c/k})$ error term from Theorem A.2.

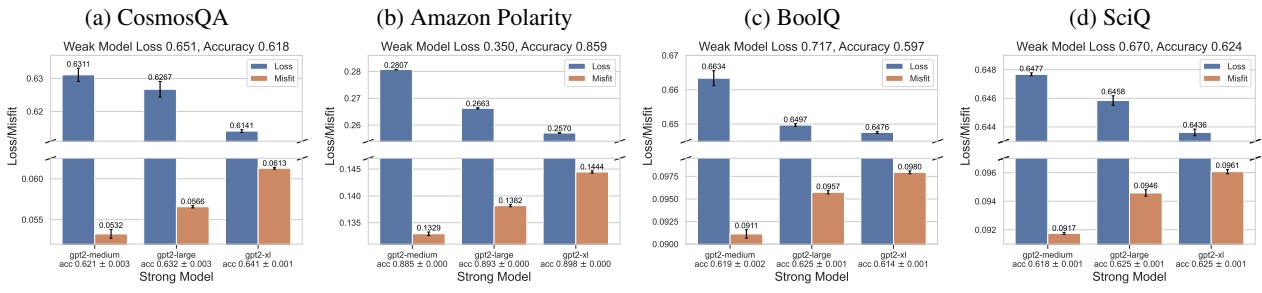

Figure 2. Weak model (gpt2) is trained once on true labels and thereafter fixed. Each strong model is trained on the weak labels (on a held-out set separate from that which weak model was trained on) for 10 random initializations of the $k$-convex combination of logistic regression heads; we plot the average test loss/misfit across these 10 runs, along with the standard deviations as the error bars.

of the linear probe: while Burns et al. (2024) adopt standard cross-entropy minimization on a single logistic head, we employ the reverse KL minimization on a convex combination of $k$ heads, as suggested by our theory.

### 5.4. Varying $k$

Recall that with a convex combination of $k$ logistic heads, the upper bound on the difference between misfit and gain in Theorem A.2 scales as $O(\sqrt{c/k})$; in particular, as $k$ increases, the upper bound becomes smaller. As our next experiment, for each dataset, we fix the strong model to be the largest one (gpt2-xl for NLP, and ViT-B/8 for Vision), and vary $k$ in $\{1, 10, 50, 100\}$. For each dataset, we plot the difference between misfit and gain against $k$; the plots are shown in Figure 4 in Appendix A.4. We observe that for all datasets but ImageNet, the difference between misfit and gain consistently decreases as $k$ increases; furthermore, beyond a point, increasing $k$ does not decrease the discrepancy by much. This can also be seen in Figure 3c, which collates the plots for all datasets (except ImageNet) in a single figure. We suspect that the trend does not show up for ImageNet because $c = 1000$, and hence $c$ dominates in the error term for the values of $k$ that we consider.

We chose to not include ImageNet in this plot as the plot for ImageNet was significantly off scale, and was skewing the y-axis.

### 5.5. Enforcing the Realizability Assumption

Finally, while the plots in Figure 2 and Figure 3a, 3b do illustrate that the gain in performance is directly proportional to the misfit as expected, our theory would also additionally suggest that the gain should *quantitatively* be at least the misfit (up to an error term decreasing in $k$). This does not quite hold in the plots—we can observe that the misfit is consistently larger than the difference between the weak model and strong model loss. One significant reason for this is that our result assumes realizability; namely, the target task should be *exactly* representable by the strong model. We verified that this does not actually hold in our experiments—even if we train the strong models on data with ground-truth labels, we see a non-trivial test loss at the end of training. To isolate this cause of discrepancy, we can consider evaluating the test losses of the weak and strong models (two terms on the LHS of equation 7) with respect to the best possible strong model (that is trained on true labels) instead of the ground-truth target function. This simply ensures that the realizability assumption holds. If we evaluate the quantities thus, consolidate the numbers across all the different NLP and vision datasets, and plot them with axes as in Figure 1, we obtain Figure 3d. Note how this plot is more aligned to the plots in Figure 1, with the misfit more faithfully capturing the quantitative gain in performance.

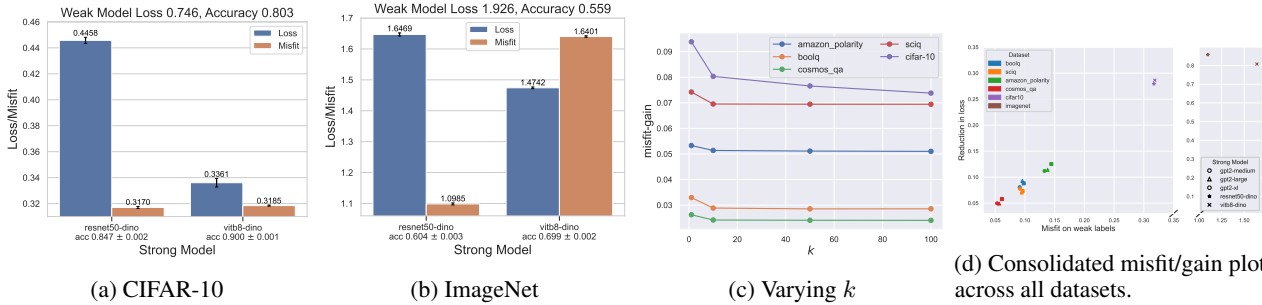

(a) CIFAR-10      (b) ImageNet      (c) Varying $k$      (d) Consolidated misfit/gain plot across all datasets.

*Figure 3.* (a), (b) Weak model (AlexNet) is trained once on true labels and thereafter fixed; we report numbers averaged over 10 runs of weak-to-strong training. (c) We observe that the difference between misfit and gain decreases as $k$ increases, and also that the decrease slows down with increasing $k$. (d) The test loss of the weak and strong model is measured not with respect to the ground truth test data, but instead with respect to the predictions of the best strong model on the test data. This is done to ensure realizability.

| | CosmosQA | | Amazon Polarity | | BoolQ | | SciQ | | CIFAR-10 | | ImageNet | |
|---|---|---|---|---|---|---|---|---|---|---|---|---|
| | Forward | Reverse | Forward | Reverse | Forward | Reverse | Forward | Reverse | Forward | Reverse | Forward | Reverse |
| Test Accuracy | 0.6378 | **0.6407** | 0.8963 | **0.8984** | 0.6140 | **0.6141** | 0.6171 | **0.6254** | 0.8981 | **0.9005** | **0.7133** | 0.6986 |
| XE(gt, strong) | 0.6147 | **0.6141** | 0.2788 | **0.2570** | **0.6464** | 0.6476 | 0.6439 | **0.6436** | 0.4052 | **0.3361** | 1.4042 | 1.4742 |
| Gain | 0.0368 | 0.0372 | 0.0717 | 0.0934 | 0.0706 | 0.0694 | 0.0264 | 0.0266 | 0.1772 | 0.2447 | 0.5217 | 0.4517 |
| Misfit | 0.6093 | 0.0613 | 0.3717 | 0.1444 | 0.6319 | 0.0980 | 0.6558 | 0.0961 | 1.1024 | 0.3185 | 3.9392 | 1.6401 |

*Table 1.* Comparison of Forward XE and Reverse KL.

## 5.6. Comparing Forward XE and Reverse KL

Finally, we empirically compare the performance of the forward and reverse KL loss in weak-to-strong supervision. For the experiments in Section 5.2, in the weak-to-strong training phase, instead of finetuning a convex combination of $k = 100$ logistic regression heads on the reverse KL divergence objective (i.e., $D_{KL}(\text{strong}, \text{weak})$ in Equation (6)), we instead finetune these on the forward cross-entropy loss (i.e., $\text{XE}(\text{weak}, \text{strong})$) with the same $\ell_2$ regularization. The latter is the more standard form of finetuning, while the former arises from our theory. Table 1 shows the comparison for the test accuracy of the strong model, as well the cross-entropy loss between the ground truth and the strong model. The comparison is indeed quite interesting—we can see that the strong model that is finetuned on the reverse KL objective shows *better* final test accuracy for nearly all the datasets! Namely, we do not see significant performance degradation (in fact, we see improvement in nearly all cases) with reverse KL compared to the standard setup.

We also computed the gain and misfit terms in the Pythagorean inequality equation 7, where for the reverse KL experiment, we compute the reverse misfit that we propose (i.e., $D_{KL}(\text{strong}, \text{weak})$), whereas in the standard forward XE experiment, we compute the misfit as $\text{XE}(\text{weak}, \text{strong})$ — the "natural" misfit that one might consider. We see in the standard setup that the Pythagorean inequality is completely off (gain and misfit don't quantitatively align), whereas the reverse misfit is more represen-

tative of the gain, as confirmed by our theory. Again, this indicates a clear "directionaliy" in the Pythagorean inequality for weak-to-strong generalization in the classification setting! Note that it is plausible that in practice, running standard forward $\text{XE}(\text{weak}, \text{strong})$ minimization might lead to a minimizer that is close to the minimizer of the reverse $D_{KL}(\text{strong}, \text{weak})$.

## 6. Conclusion and Future Work

The theory in Section 4 establishes a connection between misfit of the strong learner to its weak supervisor, and the generalization gain of the strong learner over its weak teacher. Experiments in Section 5 confirm this connection. However, these results raise several questions. Firstly, the empirical results seem tighter than expected. Can the error term in Theorem 4.3 be improved, thereby more accurately explaining the empirical results? Secondly, can we use our geometric insights to improve weak-to-strong generalization beyond the standard framework? Finally, while our experiments focused on linear probes (Burns et al., 2024), Theorem 4.3 applies to significantly larger classes of models. How tightly does Theorem 4.3 hold in these different settings, and can the ideas from this work be applied there to encourage better alignment?

The seminal work of Burns et al. (2024) exhibited the phenomenon of weak-to-strong generalization. We hope our work sheds some light on the mechanisms at play and provides theoretically justified modifications to the framework.

## Acknowledgements

We thank ICML reviewers for their helpful comments. This work is supported by Moses Charikar's and Gregory Valiant's Simons Investigator Awards. This work is also supported by NSF under awards 1705058 and 1934915.

## Impact Statement

This paper presents work whose goal is to advance the field of Machine Learning. There are many potential societal consequences of our work, none which we feel must be specifically highlighted here.

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

# A. Appendix

## A.1. Common Bregman Divergences and their Generator Functions

| Name | Generator | Divergence | Dual Map | Legendre Dual |
|------|-----------|------------|----------|---------------|
| L2 Loss | $\psi(x) = \frac{1}{2}x^2$ | $D_\psi(x, y) = \frac{1}{2}(x-y)^2$ | $x^* = x$ | $\psi^*(x^*) = \frac{1}{2}(x^*)^2$ |
| KL Divergence | $\psi(p) = -H(p)$ | $D_\psi(p, q) = D_{\mathrm{KL}}(p\|q)$ | $p^* = \sigma^{-1}(p)$ | $\psi^*(p^*) = \log(1 + e^{p^*})$ |
| Logistic | $\psi(x) = \log(1 + e^x)$ | $D_\psi(x, y) = D_{\mathrm{KL}}(\sigma(y)\|\sigma(x))$ | $x^* = \sigma(x)$ | $\psi^*(x^*) = -H(x^*)$ |
| Itakura-Saito | $\psi(x) = -\log x$ | $D_\psi(x, y) = \frac{x}{y} - \log\frac{x}{y} - 1$ | $x^* = 1/x$ | $\psi^*(x^*) = 1 - \log x^*$ |

*Table 2.* Examples of different univariate generator functions and their Bregman divergences.

## A.2. Generalizing the Misfit Inequality to Multiple Classes

Here we expand on the modifications needed to generalize the results discussed in Section 4 to the multi-class setting. There is both a *coordinate-dependent* and *coordinate-independent* approach to generalizing the misfit inequality. Below we detail the coordinate-dependent approach as it is easier to understand. Later we discuss the coordinate-independent approach which is more in line with the approach taken by softmax regression.

Again we make the simplifying assumption that our input data $X \sim P_X$ has finite support $\mathcal{X}$.

### A.2.1. MULTI-CLASS KL DIVERGENCE AS A BREGMAN DIVERGENCE

In the binary-classification setting, we are learning models that output a probability distribution on $\{0, 1\}$ given the input $X$. While such a distribution is written with two probability values $p$ and $1 - p$ in $[0, 1]$, it suffices to only know one of the two probability values to determine the other. Thus we can represent our space of models on data $X$ by the (finite-dimensional) function space $[0, 1]^{\mathcal{X}}$. This motivated framing KL-Divergence as a Bregman divergence generated by the *univariate* convex function $\psi(p) := p \log p + (1 - p) \log(1 - p)$. In binary-classification, we tend to interpret the output of a model $f \in [0, 1]^{\mathcal{X}}$ as the probability of the positive class; however, this choice was largely arbitrary. This arbitrariness is the *coordinate-dependence* of this formulation. However, had we chosen the output of $f : \mathcal{X} \to [0, 1]$ to be the probability of the negative class, we would have obtained the same Bregman divergence by the symmetry of $\psi$. The rigidity of the Bregman divergence to this arbitrariness is the *coordinate-independence* of the underlying theory.

In the $c$-class setting for $c > 2$, positive probability distributions are represented by $c$-dimensional vectors $p \in \Delta^{c-1}$, where $\Delta^{c-1}$ is the (open) $c - 1$-dimensional probability simplex:

$$\Delta^{c-1} := \left\{ p \in (0, 1)^c : \sum_{i=1}^c p_i = 1 \right\}.$$

Now, for $p \in \Delta^{c-1}$, knowing $c - 1$ of the $c$ probabilities is sufficient to determine the $c^{th}$ probability. Making the arbitrary choice of dropping the redundant $c$-th coordinate, we get the set

$$\blacktriangle^{c-1} := \left\{ p \in (0, 1)^{c-1} : \sum_{i=1}^{c-1} p_i < 1 \right\},$$

parameterizing positive probability distributions on $[c]$. Thus, we can represent our space of models on data $X$ by the (finite-dimensional) function space $\left\{ f : \mathcal{X} \to \blacktriangle^{c-1} \right\}$.

Using this parameterization, we can express the negative Shannon entropy as

$$\psi(p) := p_1 \log p_1 + \cdots p_{c-1} \log p_{c-1} + \left( 1 - \sum_{i=1}^{c-1} p_i \right) \log \left( 1 - \sum_{i=1}^{c-1} p_i \right).$$

Its dual map is the logit function $\sigma^{-1} : \blacktriangle^{c-1} \to \mathbb{R}^{c-1}$ with respect to the $c$-th probability

$$p^* := \sigma^{-1}(p) = \left( \log \frac{p_i}{p_c} \right)_{i=1}^{c-1},$$

where we abuse notation slightly for consistency with the binary class case. Then, we can compute the Legendre dual of $\psi$ as

$$\psi^*(p^*) = \log\left(\sum_{i=1}^{c} e^{p_i^*}\right),$$

where $p_c^* := 0$. $\psi^*$ is the *log-sum-exp* function, and its gradient is the softmax function $\sigma$. So similar to the binary case, softmax regression learns models in the dual (logit) space $\mathcal{X} \to \mathbb{R}^{c-1}$, and applies $\sigma$ to convert the outputs to primal (probability) values.

Since $\psi$ is strictly convex and $C^1$ on $\blacktriangle^{c-1}$, it determines a Bregman divergence $D_\psi(p,q) := \psi(p) - \psi(q) - \langle p - q, \nabla\psi(q)\rangle$ for $p, q \in \blacktriangle^{c-1}$. Substituting our choice of $\psi$, we get that $D_\psi(p,q)$ is the multi-class KL divergence:

$$D_\psi(p,q) = \sum_{i=1}^{c} p_i \log\frac{p_i}{q_i} = D_{\mathrm{KL}}(p\|q),$$

taking $p_c := 1 - \sum_{i=1}^{c-1} p_i$ and $q_c := 1 - \sum_{i=1}^{c-1} q_i$. Using this Bregman divergence, we can state the multi-class misfit analogs of the results Corollary 4.2 and Theorem 4.3 in Section 4. We first include a proof of Theorem 4.1 in more detail than in the main body of the paper.

**Theorem 4.1.** *Let $\psi : \mathbb{R}^n \to \overline{\mathbb{R}}^+$ be a proper convex function s.t. $U_\psi \neq \emptyset$. Let $h_s : \mathcal{X} \to \mathbb{R}^{d_s}$ and $h_w : \mathcal{X} \to \mathbb{R}^{d_w}$ be the strong and weak learner representations respectively. Let $f_w : \mathbb{R}^{d_w} \to U_\psi$ be the weak model finetune layer, and $g : \mathcal{X} \to U_\psi$ be the target function. Let $\mathcal{F}$ be a class of functions mapping $\mathbb{R}^{d_s} \to U_\psi$. If the following hold:*

1. *(Realizability) $\exists f_* \in \mathcal{F}$ s.t. $g = f_* \circ h_s$,*

2. *(Convexity) $\mathcal{F}$ is a convex set of functions,*

3. *(Sequential Consistency) For $y \in U_\psi$ fixed, if $D_\psi(x_n, y) \to 0$, then $x_n \to y$.*

*Then for any $\epsilon > 0$, there exists $\delta > 0$ such that for all $f_s \in \mathcal{F}$ that satisfy*

$$\mathbb{E}_X\left[D_\psi\left(f_s(h_s(X)), f_w(h_w(X))\right)\right] \leq \inf_{f \in \mathcal{F}} \mathbb{E}_X\left[D_\psi\left(f(h_s(X)), f_w(h_w(X))\right)\right] + \delta,$$

*we have*

$$\mathbb{E}_X\left[D_\psi\left(g(X), f_s(h_s(X))\right)\right] \leq \mathbb{E}_X\left[D_\psi\left(g(X), f_w(h_w(X))\right)\right] - \mathbb{E}_X\left[D_\psi\left(f_s(h_s(X)), f_w(h_w(X))\right)\right] + \epsilon. \tag{8}$$

*Proof.* First note that convexity of $\mathcal{F}$ implies convexity of $\mathcal{F} \circ h_s := \{f \circ h_s : f \in \mathcal{F}\}$: for any $f_1, f_2 \in \mathcal{F}$, $\lambda \in [0,1]$, and $x \in \mathcal{X}$, we have

$$\lambda f_1(h_s(x)) + (1-\lambda) f_2(h_s(x)) = (\lambda f_1 + (1-\lambda) f_2)(h_s(x)) \in \mathcal{F} \circ h_s.$$

Since $g \in \overline{\mathcal{F} \circ h_s}$, by Fact 3.4, we have that $g_{\mathrm{proj}} := P_{\overline{\mathcal{F} \circ h_s}}(f_w \circ h_w)$ uniquely exists and

$$\mathbb{E}_X\left[D_\psi(g(X), g_{\mathrm{proj}}(X))\right] \leq \mathbb{E}_X\left[D_\psi(g(X), f_w(h_w(X)))\right] - \mathbb{E}_X\left[D_\psi(g_{\mathrm{proj}}(X), f_w(h_w(X)))\right].$$

Now we approximate $g_{\mathrm{proj}}$ sufficiently well using $\mathcal{F} \circ h_s$. By continuity of both $\mathbb{E}_X\left[D_\psi(\cdot, f_w(h_w(X)))\right]$ and $\mathbb{E}_X\left[D_\psi(g(X), \cdot)\right]$, there exists $\epsilon_1 > 0$ s.t. if $\mathbb{E}_X\left[\|g(X) - f_s(h_s(X))\|_2^2\right] \leq \epsilon_1$, then

$$\mathbb{E}_X\left[D_\psi(g(X), f_s(h_s(X)))\right] \leq \mathbb{E}_X\left[D_\psi(g(X), f_w(h_w(X)))\right] - \mathbb{E}_X\left[D_\psi(f_s(h_s(X)), f_w(h_w(X)))\right] + \epsilon.$$

By sequential consistency, there exists $\delta > 0$ s.t. if $x, y \in U_\psi$ are s.t. $D_\psi(x, y) \leq \delta$, then $\|x - y\|_2^2 \leq \epsilon_1$. Let $p_{\min} := \min_{x \in \mathcal{X}} P_X(x) > 0$. Let $f_s \in \mathcal{F}$ be s.t.

$$\mathbb{E}_X\left[D_\psi(f_s(h_s(X)), f_w(h_w(X)))\right] \leq \inf_{f \in \mathcal{F}} \mathbb{E}_X\left[D_\psi(f(h_s(X)), f_w(h_w(X)))\right] + p_{\min}\delta.$$

By definition of $g_{\text{proj}}$, we have that

$$\mathbb{E}_X\left[D_\psi(g_{\text{proj}}(X), f_w(h_w(X)))\right] = \min_{g_2 \in \overline{\mathcal{F} \circ h_s}} \mathbb{E}_X\left[D_\psi(g_2(X), f_w(h_w(X)))\right]$$
$$= \inf_{f \in \mathcal{F}} \mathbb{E}_X\left[D_\psi(f(h_s(X)), f_w(h_w(X)))\right],$$

where the last equality follows by continuity of $\mathbb{E}_X\left[D_\psi(\cdot, f_w(h_w(X)))\right]$. Then, applying Fact 3.4 again, this time to $f_s \circ h_s \in \overline{\mathcal{F} \circ h_s}$ instead of $g$, we have that

$$\mathbb{E}_X\left[D_\psi(f_s(h_s(X)), g_{\text{proj}}(X))\right] \le \mathbb{E}_X\left[D_\psi(f_s(h_s(X)), f_w(h_w(X)))\right] - \mathbb{E}_X\left[D_\psi(g_{\text{proj}}(X), f_w(h_w(X)))\right] \le p_{\min}\delta.$$

Then, for all $x \in \mathcal{X}$, $D_\psi(f_s(h_s(x)), g_{\text{proj}}(x)) \le \delta$. Thus,

$$\epsilon_1 \ge \sup_{x \in \mathcal{X}}\left[\|f_s(h_s(x)) - g_{\text{proj}}(x)\|_2^2\right]$$
$$\ge \mathbb{E}_X\left[\|f_s(h_s(X)) - g_{\text{proj}}(X)\|_2^2\right].$$

Thus,

$$\mathbb{E}_X\left[D_\psi(g(X), f_s(h_s(X)))\right] \le \mathbb{E}_X\left[D_\psi(g(X), f_w(h_w(X)))\right] - \mathbb{E}_X\left[D_\psi(f_s(h_s(X)), f_w(h_w(X)))\right] + \epsilon.$$

$\square$

Like in the binary case, the multi-class misfit-gain inequality follows as a corollary to Theorem 4.1.

**Corollary A.1** (Multi-Class Misfit-Gain Inequality)**.** *Suppose $h_s : \mathcal{X} \to \mathbb{R}^{d_s}$ is the strong learner hidden representation and $h_w : \mathcal{X} \to \mathbb{R}^{d_w}$ is the weak learner representation. Let $f_w : \mathbb{R}^{d_w} \to \blacktriangle^{c-1}$ be the classifier for the weak model, and let $g : \mathcal{X} \to \Delta^{c-1}$ be the target function. Let $\mathcal{F}$ be a class of functions mapping $\mathbb{R}^{d_s} \to \blacktriangle^{c-1}$. If the following hold:*

1. *(Realizability) $\exists f_* \in \mathcal{F}$ so that $g = f_* \circ h_s$,*

2. *(Convexity) $\mathcal{F}$ is a convex set of functions,*

*then for any $\epsilon > 0$, there exists $\delta > 0$ so that for all $f_s \in \mathcal{F}$ such that*

$$\mathbb{E}_X\left[D_{\text{KL}}(f_s(h_s(X))\|f_w(h_w(X)))\right] \le \inf_{f \in \mathcal{F}} \mathbb{E}_X\left[D_{\text{KL}}(f(h_s(X))\|f_w(h_w(X)))\right] + \delta,$$

*we have*

$$\mathbb{E}_X\left[\text{XE}(g(X)\|f_s(h_s(X)))\right] \le \mathbb{E}_X\left[\text{XE}(g(X)\|f_w(h_w(X)))\right] - \mathbb{E}_X\left[D_{\text{KL}}(f_s(h_s(X))\|f_w(h_w(X)))\right] + \epsilon. \quad (9)$$

*Proof.* First note that by subtracting $\mathbb{E}_X\left[H(g(X))\right]$ from both sides of equation 9, we get

$$\mathbb{E}_X\left[D_{\text{KL}}(g(X)\|f_s(h_s(X)))\right] \le \mathbb{E}_X\left[D_{\text{KL}}(g(X)\|f_w(h_w(X)))\right] - \mathbb{E}_X\left[D_{\text{KL}}(f_s(h_s(X))\|f_w(h_w(X)))\right] + \epsilon,$$

so it suffices to prove this latter inequality. But this inequality follows from Theorem 4.1 after noting $D_{\text{KL}}$ is sequentially consistent by Pinsker's inequality. $\square$

### A.2.2. PROOF OF THEOREM 4.3 FOR MULTIPLE CLASSES

Next we prove Theorem 4.3 for the multi-class setting. From this result we will recover the Theorem 4.3 for binary classification. Recall that for a set $S$ in a vector space $V$, $\text{co}^k S$ is defined as

$$\text{co}^k S := \left\{ y \in V : \exists x_1, \dots, x_k \in S, p \in \Delta^{k-1} \text{ s.t. } y = \sum_{i=1}^k p_i x_i \right\}.$$

**Theorem A.2.** *Let $h_s, h_w, f_w, g$ be as in Corollary A.1. Let $\mathcal{F}$ be a class of functions mapping $\mathbb{R}^{d_s} \to \blacktriangle^{c-1}$. If the following hold:*

1. *(Realizability)* $\exists f_* \in \mathcal{F}$ *so that* $g = f_* \circ h_s$,

2. *(Regularization)* $\mathcal{F}$ *is s.t.* $\max_{i \in [c]} \mathbb{E}_X \left[ \sup_{f \in \mathcal{F}} 1/f_i(X) \right] < \infty$.

*Then for any* $k \in \mathbb{N}$, *there exists* $\delta > 0$ *so for all* $f_s \in \mathrm{co}^k \mathcal{F}$ *such that*

$$\mathbb{E}_X \left[ D_{\mathrm{KL}}(f_s(h_s(X)) \| f_w(h_w(X))) \right] \leq \inf_{f \in \mathrm{co}^k \mathcal{F}} \mathbb{E}_X \left[ D_{\mathrm{KL}}(f(h_s(X)) \| f_w(h_w(X))) \right] + \delta,$$

*we have*

$$\mathbb{E}_X \left[ \mathrm{XE}(g(X) \| f_s(h_s(X))) \right] \leq \mathbb{E}_X \left[ \mathrm{XE}(g(X) \| f_w(h_w(X))) \right] - \mathbb{E}_X \left[ D_{\mathrm{KL}}(f_s(h_s(X)) \| f_w(h_w(X))) \right] + O(\sqrt{\tfrac{c}{k}}). \quad (10)$$

The proof will come down to determining how large we need to make $k$ to get satisfactory bounds on both

$$\mathbb{E}_X \left[ D_{\mathrm{KL}}(g_{\mathrm{proj}}(X) \| f_w(h_w(X))) \right] - \mathbb{E}_X \left[ D_{\mathrm{KL}}(f_s(h_s(X)) \| f_w(h_w(X))) \right],$$
$$\mathbb{E}_X \left[ D_{\mathrm{KL}}(g(X) \| g_{\mathrm{proj}}(X)) \right] - \mathbb{E}_X \left[ D_{\mathrm{KL}}(g(X) \| f_s(h_s(X))) \right],$$

where $g_{\mathrm{proj}} := P_{\overline{\mathrm{co}}(\mathcal{F} \circ h_s)}(f_w \circ h_w)$. We will see the first of the two terms is controlled by the *Jensen approximation gap* of $\psi$ for a Bregman divergence $D_\psi$. The second term is more difficult to bound from general Bregman theory. We will instead bound it in specifically for KL divergence, leaving it open what conditions are necessary to get a satisfactory bound for general Bregman divergences.

For a random variable $Z \sim P_Z$, we let $\overline{Z}^{(n)}$ denote the empirical mean of $n$ i.i.d. samples of $Z$.

**Definition A.3.** Let $\varphi : \mathbb{R}^d \to \overline{\mathbb{R}}^+$ be a proper, convex function. The *Jensen approximation gap* of $\varphi$ is

$$\mathrm{Gap}(n; \varphi) := \sup_{\substack{Z \sim P_Z \text{ on } \mathrm{dom}\psi \\ \mathbb{E}Z \in \mathrm{dom}\psi}} \mathbb{E} \left[ \varphi(\overline{Z}^{(n)}) \right] - \varphi(\mathbb{E}[Z]).$$

**Lemma A.4.** *Let* $\psi : \mathbb{R}^d \to \overline{\mathbb{R}}^+$ *be the generator for Bregman divergence* $D_\psi$. *Let* $S \subset U_\psi$ *and let* $w \in U_\psi$. *Then for any* $k \in \mathbb{N}$ *and* $y \in \mathrm{co}S$, *we have that there exists* $x \in \mathrm{co}^k S$ *s.t.*

$$D_\psi(x, w) \leq D_\psi(y, w) + \mathrm{Gap}(k; \psi).$$

Note that in the finite-dimensional case Caratheodory's theorem does tell us that $\mathrm{co}^{d+1}S = \mathrm{co}S$. However, we should imagine $d \gg k$. In the infinite-dimensional case, Caratheodory's theorem does not hold, so we need to use the approach from the lemma above. The proof of the lemma applies probabilistic method in a way similar to (Zeevi & Meir, 1997).

*Proof.* Let $y \in \mathrm{co}S$. Then there exist $n \in \mathbb{N}$, $z_1, \ldots, z_n \in S$, and $(p_1, \ldots, p_n) \in \Delta^{n-1}$ s.t. $y = \sum_{i=1}^n p_i z_i$. Note that $p$ is a probability distribution on $[n]$. Let $Z \sim P_Z$ be a random variable on $\{z_1, \ldots, z_n\}$ with $P_Z(z_i) := p_i$. Then $\mathbb{E}Z = y$. Now observe

$$\mathbb{E}D_\psi(\overline{Z}^{(k)}, w) - D_\psi(y, w) = \mathbb{E} \left[ \psi(\overline{Z}^{(k)}) - \psi(w) - \langle \overline{Z}^{(k)} - w, \nabla\psi(w) \rangle \right] - \psi(y) + \psi(w) + \langle y - w, \nabla\psi(w) \rangle$$

$$= \mathbb{E} \left[ \psi(\overline{Z}^{(k)}) \right] - \langle \mathbb{E}\overline{Z}^{(k)}, \nabla\psi(w) \rangle - \psi(y) + \langle y, \nabla\psi(w) \rangle$$

$$= \mathbb{E} \left[ \psi(\overline{Z}^{(k)}) \right] - \psi(y)$$

$$\leq \mathrm{Gap}(k; \psi).$$

So there exist $i_1, \ldots, i_k \in [n]$ and $x := \frac{1}{k} \sum_{j=1}^k z_{i_j} \in \mathrm{co}^k S$ s.t.

$$D_\psi(x, w) - D_\psi(y, w) \leq \mathbb{E}D_\psi(\overline{Z}^{(k)}, w) - D_\psi(y, w) \leq \mathrm{Gap}(k; \psi).$$

$\square$

From here, we now turn to bounding the Jensen approximation gap of the negative Shannon entropy function. While there is a extensive literature on bounding Jensen gaps (Abramovich & Persson, 2016; Gao et al., 2020; Ullah et al., 2021; Konenkov, 2024), we find that a simple method inspired by (https://math.stackexchange.com/users/955195/small deviation) is sufficient to get our desired bound.

**Lemma A.5.** *Let $S \subset \blacktriangle^{c-1}$ be a set s.t. $a := (\sup_{p \in S}(1/p_i))_{i=1}^c$ is finite in all its coordinates (here $p_c := 1 - \sum_{i=1}^{c-1} p_i$). Let $\psi : \blacktriangle^{c-1} \to \overline{\mathbb{R}}^+$ be the negative Shannon entropy function and let $\psi|_{coS}$ be $\psi$ restricted to $coS$. Then for any $k \in \mathbb{N}$, we have*

$$\mathrm{Gap}(k; \psi|_{coS}) \leq \frac{1}{8k} \sum_{i=1}^c a_i \left( 1 - \sum_{j=1}^c \frac{1}{a_j} \right)^2 \leq \frac{1}{8k} \sum_{i=1}^c a_i.$$

*Proof.* First note that it suffices to assume $S$ is convex. Indeed for any $q \in coS$, note that $a_i \geq \frac{1}{q_i}$ for all $i \in [c]$, since $x \mapsto \frac{1}{x}$ is convex.

So assume $S$ is convex. It is more convenient to consider the extension of $\psi$ to all $\mathbb{R}_{>0}^c := \{x \in \mathbb{R}^c : x_i > 0 \ \forall i \in [c]\}$. We call this extension $\widetilde{\psi} : \mathbb{R}_{>0}^c \to \mathbb{R}$ and define it as

$$\widetilde{\psi}(x) := \sum_{i=1}^c x_i \log x_i.$$

Let $\iota : \blacktriangle^{c-1} \to \Delta^{c-1}$ be the map defined

$$\iota(p_1, \ldots, p_{c-1}) := (p_1, \ldots, p_{c-1}, 1 - \sum_{i=1}^{c-1} p_i).$$

Then $\psi|_S = \widetilde{\psi} \circ \iota|_S$. So $\mathrm{Gap}(k; \psi|_S) = \mathrm{Gap}(k; \widetilde{\psi} \circ \iota|_S)$, and it suffices to bound the Jensen approximation gap of $\widetilde{\psi}$ on $\iota(S)$.

To that end, let $Z \sim P_Z$ be a random variable on $\iota(S)$. Since $\iota(S)$ is bounded and convex, $\mathbb{E}Z$ is well-defined and in $\iota(S)$. Let $\overline{Z}^{(k)}$ be the empirical mean of $k$ i.i.d. samples drawn from $P_Z$. Note that $(\nabla^2 \widetilde{\psi}(x))_{ij} = \delta_{ij} \frac{1}{x_i}$ for $i, j \in [c]$, where $\delta_{ij}$ is the Kroenecker delta. So $\widetilde{\psi}(x) - \frac{1}{2} \sum_{i=1}^c a_i x_i^2$ is concave as a function of $x \in \iota(S)$. By Jensen's inequality, we have

$$\mathbb{E}\left[ \widetilde{\psi}(\overline{Z}^{(k)}) - \frac{1}{2} \sum_{i=1}^c a_i (\overline{Z}_i^{(k)})^2 \right] \leq \widetilde{\psi}(\mathbb{E}\overline{Z}^{(k)}) - \frac{1}{2} \sum_{i=1}^c a_i (\mathbb{E}\overline{Z}_i^{(k)})^2$$

$$\Rightarrow \mathbb{E}\left[ \widetilde{\psi}(\overline{Z}^{(k)}) \right] - \widetilde{\psi}(\mathbb{E}\overline{Z}^{(k)}) \leq \frac{1}{2} \sum_{i=1}^c a_i \left( \mathbb{E}\left[ (\overline{Z}_i^{(k)})^2 \right] - (\mathbb{E}Z_i)^2 \right)$$

$$\Rightarrow \mathbb{E}\left[ \widetilde{\psi}(\overline{Z}^{(k)}) \right] - \widetilde{\psi}(\mathbb{E}\overline{Z}^{(k)}) \leq \frac{1}{2k} \sum_{i=1}^c a_i \mathrm{Var}\left[ Z_i \right].$$

Since each $Z_i \in (1/a_i, 1 - \sum_{j=1, j \neq i}^c 1/a_j)$, Popoviciu's inequality tells us

$$\mathrm{Var}\left[ Z_i \right] \leq \frac{1}{4} \left( 1 - \sum_{j=1, j \neq i}^c \frac{1}{a_j} - \frac{1}{a_i} \right)^2 = \frac{1}{4} \left( 1 - \sum_{j=1}^c \frac{1}{a_j} \right)^2 \leq \frac{1}{4}.$$

As $Z$ was arbitrary, we have that

$$\mathrm{Gap}(k; \psi|_S) = \mathrm{Gap}(k; \widetilde{\psi} \circ \iota|_S) \leq \frac{1}{8k} \sum_{i=1}^c a_i \left( 1 - \sum_{j=1}^c \frac{1}{a_i} \right)^2 \leq \frac{1}{8k} \sum_{i=1}^c a_i.$$

$\square$

As a corollary, we bound the "functional" Jensen approximation gap:

**Corollary A.6.** *Let $X \sim P_X$ be a random variable on finite set $\mathcal{X}$. Let $\mathcal{F} \subset (\blacktriangle^{c-1})^{\mathcal{X}}$ be a class of functions s.t. $\rho_i := \mathbb{E}_X \left[ \sup_{f \in \mathcal{F}} (1/f_i(X)) \right] < \infty$ for all $i \in [c]$ (where $f_c := 1 - \sum_{i=1}^{c-1} f_i$). Let $\psi : \blacktriangle^{c-1} \to \overline{\mathbb{R}}^+$ be the negative Shannon entropy function and let $\Psi : (\blacktriangle^{c-1})^{\mathcal{X}} \to \overline{\mathbb{R}}^+$ be defined $\Psi[f] := \mathbb{E}_X [\psi(f(X))]$. Let $\Psi\big|_{\mathrm{co}\mathcal{F}}$ be $\Psi$ restricted to $\mathrm{co}\mathcal{F}$. Then for any $k \in \mathbb{N}$, we have*

$$\mathrm{Gap}(k; \Psi\big|_{\mathrm{co}\mathcal{F}}) \le \frac{1}{8k} \sum_{i=1}^{c} \rho_i.$$

*Proof.* Note that $\mathbb{E}_X \left[ \mathrm{Gap}(k; \psi\big|_{\mathcal{F}(X)}) \right] \ge \mathrm{Gap}(k; \Psi\big|_{\mathcal{F}})$. Thus,

$$\mathrm{Gap}(k; \Psi\big|_{\mathcal{F}}) \le \mathbb{E}_X \left[ \frac{1}{8k} \sum_{i=1}^{c} \sup_{f \in \mathcal{F}} 1/f_i(X) \right] = \frac{1}{8k} \sum_{i=1}^{c} \rho_i.$$

$\square$

To complete the proof of Theorem A.2, we need one more bound:

**Lemma A.7.** *Let $p, q \in \Delta^{c-1}$. Then*

$$\max_{i \in [c]} |\log p_i - \log q_i| \le \frac{\sqrt{2D_{\mathrm{KL}}(p\|q)}}{\min_{i \in [c]} (p_i, q_i)}.$$

*Proof.* Observe that

$$\max_{i \in [c]} |\log p_i - \log q_i| \le \max_{i \in [c]} \left( \max(1/p_i, 1/q_i) |p_i - q_i| \right) \qquad \text{(Mean-Value Theorem)}$$

$$\le \max_{i \in [c]} (1/p_i, 1/q_i) \sum_{i=1}^{c} |p_i - q_i|$$

$$\le \max_{i \in [c]} (1/p_i, 1/q_i) \sqrt{2D_{\mathrm{KL}}(p\|q)} \qquad \text{(Pinsker's Inequality)}$$

$$= \frac{\sqrt{2D_{\mathrm{KL}}(p\|q)}}{\min_{i \in [c]} (p_i, q_i)}.$$

$\square$

Numerical calculations we did when $c = 2$ suggest the bound in Lemma A.7 can be improved to

$$\max_{i \in [c]} |\log p_i - \log q_i| \le \sqrt{\frac{2D_{\mathrm{KL}}(p\|q)}{\min_{i \in [c]} (p_i, q_i)}}.$$

However, the dependence on $\min_{i \in [c]} (p_i, q_i)$ does appear to be necessary. We leave it for future work to tighten this bound.

With these lemmas, we can now prove the main result for the multi-class setting when our strong class is not convex.

*Proof of Theorem A.2 and 4.3.* Similarly to the proof of Corollary A.1, it suffices to prove the inequality

$$\mathbb{E}_X [D_{\mathrm{KL}}(g(X)\|f_s(h_s(X)))] \le \mathbb{E}_X [D_{\mathrm{KL}}(g(X)\|f_w(h_w(X)))] - \mathbb{E}_X [D_{\mathrm{KL}}(f_s(h_s(X))\|f_w(h_w(X)))] + O(\sqrt{\tfrac{c}{k}})$$

instead.

Let $\rho_i := \mathbb{E}_X \left[ \sup_{f \in \mathcal{F}} 1/f_i(X) \right]$ for $i \in [c]$. By Corollary A.6 and Lemma A.4, we have that for all $f^{(\mathrm{conv})} \in \mathrm{co}\mathcal{F}$,

$$\inf_{f \in \mathrm{co}^k \mathcal{F}} \mathbb{E}_X [D_{\mathrm{KL}}(f(h_s(X))\|f_w(h_w(X)))] \le \mathbb{E}_X \left[ D_{\mathrm{KL}}(f^{(\mathrm{conv})}(h_s(X))\|f_w(h_w(X))) \right] + \frac{1}{8k} \sum_{i=1}^{c} \rho_i.$$

Thus,

$$\inf_{f \in \mathrm{co}^k \mathcal{F}} \mathbb{E}_X \left[ D_{\mathrm{KL}}(f(h_s(X)) \| f_w(h_w(X))) \right] \leq \inf_{f \in \mathrm{co}\mathcal{F}} \mathbb{E}_X \left[ D_{\mathrm{KL}}(f(h_s(X)) \| f_w(h_w(X))) \right] + \frac{1}{8k} \sum_{i=1}^{c} \rho_i.$$

Let $g^{(\mathrm{proj})} := P_{\overline{\mathrm{co}}(\mathcal{F} \circ h_s)}(f_w \circ h_w)$. Since

$$\mathbb{E}_X \left[ D_{\mathrm{KL}}(g^{(\mathrm{proj})}(X) \| f_w(h_w(X))) \right] = \inf_{f \in \mathrm{co}\mathcal{F}} \mathbb{E}_X \left[ D_{\mathrm{KL}}(f(h_s(X)) \| f_w(h_w(X))) \right],$$

we have that

$$\inf_{f \in \mathrm{co}^k \mathcal{F}} \left[ \mathbb{E}_X D_{\mathrm{KL}}(f(h_s(X)) \| f_w(h_w(X))) \right] \leq \mathbb{E}_X \left[ D_{\mathrm{KL}}(g^{(\mathrm{proj})}(X) \| f_w(h_w(X))) \right] + \frac{1}{8k} \sum_{i=1}^{c} \rho_i.$$

Applying Corollary A.1, for any $\epsilon > 0$, if $f_s \in \mathrm{co}^k \mathcal{F}$ is s.t.

$$\mathbb{E}_X \left[ D_{\mathrm{KL}}(f_s(h_s(X)) \| f_w(h_w(X))) \right] \leq \inf_{f \in \mathrm{co}^k \mathcal{F}} \mathbb{E}_X \left[ D_{\mathrm{KL}}(f(h_s(X)) \| f_w(h_w(X))) \right] + \epsilon,$$

then

$$\mathbb{E}_X \left[ D_{\mathrm{KL}}(g(X) \| g^{(\mathrm{proj})}(X)) \right] \leq \mathbb{E}_X \left[ D_{\mathrm{KL}}(g(X) \| f_w(h_w(X))) \right] - \mathbb{E}_X \left[ D_{\mathrm{KL}}(g^{(\mathrm{proj})}(X) \| f_w(h_w(X))) \right]$$

$$\leq \mathbb{E}_X \left[ D_{\mathrm{KL}}(g(X) \| f_w(h_w(X))) \right] - \inf_{f \in \mathrm{co}^k \mathcal{F}} \mathbb{E}_X \left[ D_{\mathrm{KL}}(f(h_s(X)) \| f_w(h_w(X))) \right] + \frac{1}{8k} \sum_{i=1}^{c} \rho_i$$

$$\leq \mathbb{E}_X \left[ D_{\mathrm{KL}}(g(X) \| f_w(h_w(X))) \right] - \mathbb{E}_X \left[ D_{\mathrm{KL}}(f_s(h_s(X)) \| f_w(h_w(X))) \right] + \epsilon + \frac{1}{8k} \sum_{i=1}^{c} \rho_i.$$

Now it remains to bound the left-hand side. First observe that, by applying to Corollary A.1 to $f_s \circ h_s$ instead of $g$, we have

$$\mathbb{E}_X D_{\mathrm{KL}}(f_s(h_s(X)) \| g^{(\mathrm{proj})}(X)) \leq \mathbb{E}_X D_{\mathrm{KL}}(f_s(h_s(X)) \| f_w(h_w(X))) - \mathbb{E}_X D_{\mathrm{KL}}(g^{(\mathrm{proj})}(X) \| f_w(h_w(X)))$$

$$\leq \frac{1}{8k} \sum_{i=1}^{c} \rho_i + \epsilon.$$

Now we apply Lemma A.7. First note that

$$\left| \mathbb{E}_X \left[ D_{\mathrm{KL}}(g(X) \| g^{(\mathrm{proj})}(X)) \right] - \mathbb{E}_X \left[ D_{\mathrm{KL}}(g(X) \| f_s(h_s(X))) \right] \right| \leq \mathbb{E}_X \left[ \left| D_{\mathrm{KL}}(g(X) \| g^{(\mathrm{proj})}(X)) - D_{\mathrm{KL}}(g(X) \| f_s(h_s(X))) \right| \right]$$

$$\leq \mathbb{E}_X \left[ \sum_{i=1}^{c} g_i(X) |\log g_i^{(\mathrm{proj})}(X) - \log f_{s,i}(h_s(X))| \right]$$

$$\leq \mathbb{E}_X \left[ \max_{i \in [c]} |\log g_i^{(\mathrm{proj})}(X) - \log f_{s,i}(h_s(X))| \right]$$

Using Lemma A.7, we then have

$$\left| \mathbb{E}_X \left[ D_{\mathrm{KL}}(g(X) \| g^{(\mathrm{proj})}(X)) \right] - \mathbb{E}_X \left[ D_{\mathrm{KL}}(g(X) \| f_s(h_s(X))) \right] \right| \leq \mathbb{E}_X \left[ \frac{\sqrt{2 D_{\mathrm{KL}}(f_s(h_s(X)) \| g^{(\mathrm{proj})}(X))}}{\min_{i \in [c]}(f_{s,i}(X), g_i^{(\mathrm{proj})}(X))} \right]$$

$$\leq \mathbb{E}_X \left[ \max_{i \in [c]} \rho_i \sqrt{\frac{c}{4k} \max_{i \in [c]} \rho_i + \epsilon} \right]$$

$$\leq \left( \max_{i \in [c]} \rho_i \right)^{3/2} \sqrt{\frac{c}{4k}} + \epsilon_2,$$

where $\epsilon_2 \to 0$ as $\epsilon \to 0$. Incorporating this estimation into

$$\mathbb{E}_X \left[ D_{\mathrm{KL}}(g(X)\|g^{(\mathrm{proj})}(X)) \right] \leq \left[ \mathbb{E}_X D_{\mathrm{KL}}(g(X)\|f_w(h_w(X))) \right] - \mathbb{E}_X \left[ D_{\mathrm{KL}}(f_s(h_s(X))\|f_w(h_w(X))) \right] + \epsilon + \frac{1}{8k} \sum_{i=1}^{c} \rho_i,$$

we have

$$\mathbb{E}_X \left[ D_{\mathrm{KL}}(g(X)\|f_s(h_s(X))) \right] \leq \mathbb{E}_X \left[ D_{\mathrm{KL}}(g(X)\|f_w(h_w(X))) \right] - \mathbb{E}_X \left[ D_{\mathrm{KL}}(f_s(h_s(X))\|f_w(h_w(X))) \right]$$
$$+ \epsilon + \frac{1}{8k} \sum_{i=1}^{c} \rho_i + \left( \max_{i \in [c]} \rho_i \right)^{3/2} \sqrt{\frac{c}{4k}} + \epsilon_2.$$

Since we are free to choose $\epsilon$ as small as we want, we can make $\epsilon = o(1/\sqrt{k})$, ensuring that the error term is $O(\sqrt{c/k})$. Therefore, for every $k \in \mathbb{N}$, there exists $\delta > 0$ s.t. for all $f_s \in \mathrm{co}^k \mathcal{F}$ for which

$$\mathbb{E}_X \left[ D_{\mathrm{KL}}(f_s(h_s(X))\|f_w(h_w(X))) \right] \leq \inf_{f \in \mathrm{co}^k \mathcal{F}} \mathbb{E}_X \left[ D_{\mathrm{KL}}(f(h_s(X))\|f_w(h_w(X))) \right] + \delta,$$

we have

$$\mathbb{E}_X \left[ D_{\mathrm{KL}}(g(X)\|f_s(h_s(X))) \right] \leq \mathbb{E}_X \left[ D_{\mathrm{KL}}(g(X)\|f_w(h_w(X))) \right] - \mathbb{E}_X \left[ D_{\mathrm{KL}}(f_s(h_s(X))\|f_w(h_w(X))) \right] + O\left( \sqrt{c/k} \right).$$

$\square$

As noted below Lemma A.7, numerical calculations of the optimal bound for Lemma A.7 for $c = 2$ suggests that our constant term in the $O\left( \sqrt{c/k} \right)$ error term can be improved from $\dfrac{\left( \max\limits_{i \in [c]} \rho_i \right)^{3/2}}{2}$ to $\dfrac{\max\limits_{i \in [c]} \rho_i}{2}$. We leave it for future work to derive this bound rigorously, and determine if the decay rate of the error term in Corollary A.1 is asymptotically tight.

### A.2.3. A BRIEF NOTE ON THE COORDINATE-INDEPENDENT APPROACH

As noted in Appendix A.2.1, the Bregman divergence $D_{\mathrm{KL}}$ for multiple classes is derived by arbitrarily choosing to drop the redundant $c$-th class. What if we had chosen to drop the $i$-th class instead, for $i \in [c]$? We can capture the fact that the underlying theory is invariant under such a choice by the use of smooth manifolds, specifically affine manifolds. A manifold $M$ is a topological space (satisfying some additional technical properties) equipped with a collection of coordinate charts $\{(U_i, \varphi_i)\}_{i \in I}$, where $U_i$ is an open subset of $M$ and $\varphi_i : U_i \to \mathbb{R}^n$ is a topological embedding, s.t. $\{U_i\}_{i \in I}$ is an open cover of $M$. For a manifold $M$ and a pair of charts $(U_1, \varphi_1)$ and $(U_2, \varphi_2)$, for open $U_1 \subset U_2$, we call the map $\varphi_2 \circ \varphi_1^{-1} : \varphi_1(U_1 \cap U_2) \to \varphi_2(U_2)$ a transition function. A manifold is smooth if all its transition functions are smooth as functions in the sense of usual multivariate calculus. A manifold is *affine* if all its transition functions are affine maps.

We can extend the notion of convexity to an affine manifold $M$ by declaring a function $\psi : M \to \overline{\mathbb{R}}^+$ to be convex at $p \in M$ if there exists a chart $(U, \varphi)$ containing $p$ such that $\psi \circ \varphi^{-1}$ is convex as a function from $\varphi(U) \subset \mathbb{R}^n$ to $\overline{\mathbb{R}}^+$. Because all transition functions are affine, if $\psi$ is convex at $p \in M$ with respect to one chart, it is convex with respect to all charts containing $p$. Thus, our definition of convexity does not rely on a specific choice of coordinates.

One can then check that Bregman divergences can be defined in a similar way that is independent of the coordinate charts. $D_{\mathrm{KL}}$ will arise when use the affine charts $\varphi_i : \Delta^{c-1} \to \blacktriangle^{c-1}$ defined by $\varphi_i(p) := (p_1, \ldots, p_{i-1}, p_{i+1}, \ldots, p_c)$ for each $i \in [c]$. The calculations we did in the previous section were manipulating $D_{\mathrm{KL}}$ with the specific chart $\varphi_c$.

This coordinate-independent approach is employed in Information Geometry. We refer the interested reader to (Amari, 2016; Ay et al., 2017) for a more in-depth treatment of this approach.

### A.3. Generalizing beyond Input Data Distributions with Finite Support

A major simplifying assumption in this work is that the input data distribution $P_X$ has finite support. When $P_X$ has infinite support or is continuous, function spaces of models become subsets of infinite-dimensional Banach spaces, and more care needs to be taken to justify the calculations we did in the previous sections.

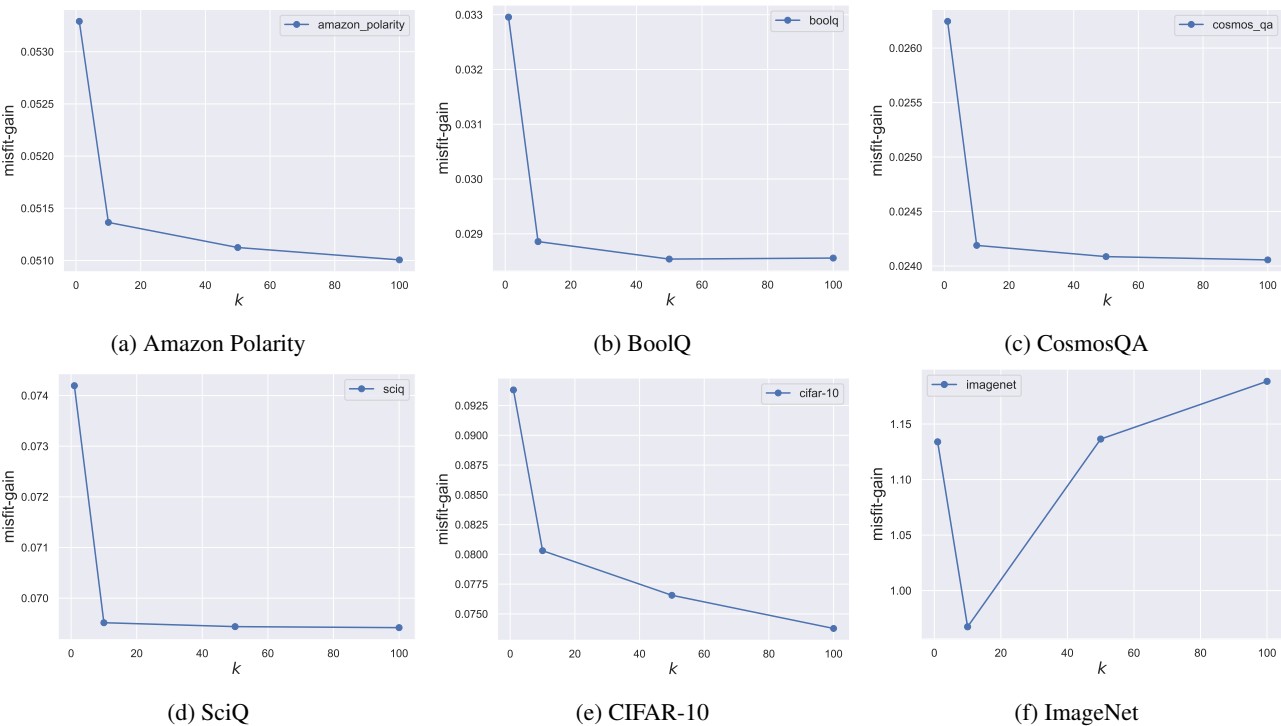

*Figure 4.* For all datasets except ImageNet, we observe that the difference between misfit and gain decreases as $k$ increases, and also that the decrease slows down with increasing $k$. Since ImageNet has $c = 1000$ classes, and we only consider values of $k$ till 100, we suspect that the $c$ term dominates in the $O(\sqrt{c/k})$ error.

The fundamentals of convex analysis in infinite-dimensional vectors spaces can be found in (Ekeland & Temam, 1999). In addition many works address the topic of Bregman divergences in infinite-dimensional spaces, with (Bauschke et al., 2003; Reem et al., 2018) being good comprehensive overviews on the topic. Below we briefly summarize the steps needed to complete the generalization.

Given an arbitrary probability measure $P_X$ on $\mathcal{X}$ let $X \sim P_X$. Suppose we have strictly convex $\psi : \mathbb{R}^d \to \overline{\mathbb{R}}^+$ that is $C^1(U_\psi)$. As described in Section 3, $\psi$ generates a Bregman divergence $D_\psi$. In Section 3.4.3, we showed that the expectation of a Bregman divergence is itself a Bregman divergence of the convex functional $\Psi[f] := \mathbb{E}_X[\psi(f(X))]$. When our function space is infinite-dimensional, we need a weaker notion of a $C^1$ function. The appropriate notion is that of a *Legendre function* defined in (Bauschke et al., 2003), extending the finite-dimensional notion from (Rockafellar, 1997). Existence and uniqueness of Bregman projections then follows from weak lower-semicontinuity of $D_\Psi$ and boundedness of the sublevel sets of $D_\Psi$. Theorem 4.1 follows in much the same way, but we need sequential consistency of $D_\Psi$ instead of just $D_\psi$. Fortunately, Pinsker's inequality still establishes that $\mathbb{E}_X D_{\mathrm{KL}}$ is sequentially consistent with respect to $L^p$ topologies on $\mathcal{X} \to \blacktriangle^{c-1}$ for $p < \infty$. This then completes the generalization Corollary A.1. Many of the details in our proof of Theorem 4.3 in Appendix A.2.2 generalize directly due to convenient properties of convex functions. For example, we can swap expectations when passing the Jensen gap of $\psi$ to a bound on the gap of $\Psi$ because the gap is always nonnegative by Jensen's theorem, and Tonelli's theorem allows us to swap the order of the expectations.

### A.4. Plots for Varying $k$

See Figure 4.

