# OpenReview forum: "Relating Misfit to Gain in Weak-to-Strong Generalization Beyond the Squared Loss"
_ICML.cc/2025/Conference — ICML 2025 poster_

### Official Review · Reviewer_fGrp · 2025-03-10

**Overall Recommendation:** 3

**Summary:**

This paper extends theoretical understanding of weak-to-strong generalization—where a strong model trained on weakly labeled data can surpass the weak model's performance—beyond regression with squared loss to general loss functions defined by Bregman divergences, including classification tasks with cross-entropy loss. It builds on prior work and generalizes their “misfit-to-gain” result by showing that for convex function classes and losses expressible via Bregman divergences, the gain in strong model performance can be quantified in terms of its disagreement (misfit) with the weak model. The paper further relaxes the convexity assumption by considering convex combinations of k strong models and shows that the misfit-gain inequality still holds up to an error term that vanishes with increasing k. Empirical validation is provided using both synthetic data and real-world NLP and vision benchmarks, demonstrating the theoretical predictions in practice.

**Claims And Evidence:**

The central claim—that performance gain in weak-to-strong generalization can be lower bounded by a misfit term (e.g., KL divergence between strong and weak models)—is clearly stated and supported by solid theoretical analysis. The authors rigorously derive this claim using generalized Pythagorean inequality for Bregman divergences and provide multiple variants (Theorems 4.1–4.3 and Corollary 4.2) to cover different settings (ideal convex classes, approximate convex combinations).

**Essential References Not Discussed:**

I am not very familiar with this field.

**Experimental Designs Or Analyses:**

The experimental design is solid. The synthetic setup mimics the theoretical conditions closely, and the real-world experiments provide practical relevance. Careful considerations are made in terms of controlling for realizability and analyzing the role of k.
One minor weakness is that the empirical loss reduction does not always match the theoretical lower bound (misfit) exactly, but the authors acknowledge this and explain it as a consequence of non-realizability or optimization difficulties.

**Methods And Evaluation Criteria:**

Evaluation is carried out on relevant synthetic setups and popular benchmarks like CIFAR-10, ImageNet, BoolQ, and CosmosQA.

**Other Comments Or Suggestions:**

1. The submission is not self-contained. Without reading the original paper for misfit, the reader can only understand this paper very vaguely.
2. It is very hard to understand the content in the conclusion section. I guess it is a wrong paragraph.

**Other Strengths And Weaknesses:**

Slight discrepancy between theoretical and empirical gain.

**Questions For Authors:**

Could you provide more intuition or visualization around the dual space projections used in your geometric arguments?

**Relation To Broader Scientific Literature:**

Not very clear. It may connect to the broader themes of co-training, disagreement-based learning.

**Theoretical Claims:**

The derivations appear mathematically sound, and the assumptions are clearly stated (e.g., realizability, convexity, sequential consistency). The theoretical results properly extend known results and are consistent with the underlying convex geometry.

---

> ### Author Rebuttal · Authors · 2025-04-01
>
> Thank you so much for reading our paper, and for your comments. We are glad that you like our work! We address your concerns ahead:
> >...submission is not self-contained
>
> While the high-level geometrical framework of viewing weak-to-strong generalization (WTSG) as “projections onto a convex space” is certainly inspired from the work of Charikar et al. 2024 (as we also generously attribute throughout our paper), we do want to emphasize that our theory in terms of Bregman divergences is a strict generalization of the results in Charikar et al. 2024. As such, their result about regression with the squared loss can be deduced as a special case of our Theorem 4.1 (as also stated in Lines 214-216). We do also set up all the preliminaries about Bregman Divergences and their geometry (Section 3) for deriving Theorem 4.1. Nevertheless, the feedback is well-taken; we will try and include a brief discussion of the Charikar et al. 2024 result in Section 3.1.
>
> >...conclusion section
>
> We will make sure to elaborate more on the future directions mentioned in the conclusion section.
>
> > Could you provide more intuition or visualization around the dual space projections used in your geometric arguments?
>
> Thank you for bringing this up. It can indeed be difficult to visualize the geometry of KL-divergence since it can behave quite differently from our familiar Euclidean geometry. Frank Nielsen has a very [nice article](https://www.ams.org//journals/notices/201803/rnoti-p321.pdf) discussing this very topic.
>
> Since we are most familiar with Euclidean geometry, it is best to appeal to those native intuitions when understanding dual space projections. More specifically, our geometric arguments require us to generalize the notion of angles so we can discuss the Pythagorean theorem in a more general setting. Bregman divergence theory gives us such a generalization, but it comes at a critical cost: asymmetry. In Euclidean geometry, we can measure angles between any two vectors by taking an inner-product. In Bregman geometry, we can only measure angles between a primal vector and a dual vector. But once we designate one vector as primal and one as dual, much of our traditional intuition from Euclidean geometry applies.
>
> When technically manipulating these expressions, we like to imagine that we are in Euclidean geometry and have access to inner products. The only restriction is we cannot swap terms in inner products or in squared $L_2$ distances. If proofs go through after taking into account these restrictions, then it is very likely they will hold in the general Bregman setting. We would be happy to include some more intuition and visualization regarding this in our final revision.

---

### Official Review · Reviewer_D6HR · 2025-03-11

**Overall Recommendation:** 3

**Summary:**

The paper characterizes the gain in weak-to-strong generalization by relating it to misfit, extending the results of Charikar et al. (2024) to general Bregman divergence. This work also weakens the condition on the strong model class, which was considered convex in Charikar et al. (2024), by allowing it to be a convex combination of functions in a general function space.

**Claims And Evidence:**

The main theorem statement is clearly presented, and both the proof in the appendix and the proof overview in the main text appear to be well-structured and understandable.

**Essential References Not Discussed:**

Almost all essential references are included. However, there are some recent references and concurrent works released after the ICML submission period that are not discussed. I suggest the authors include and discuss these references in the next revision. In particular, the discussion with [2] and [3] seems crucial as they are concurrent works.

[1] Medvedev, Marko, et al. "Weak-to-Strong Generalization Even in Random Feature Networks, Provably." *arXiv preprint arXiv:2503.02877* (2025).

[2] Yao, Wei, et al. "Understanding the Capabilities and Limitations of Weak-to-Strong Generalization." *arXiv preprint arXiv:2502.01458* (2025).

[3] Yao, Wei, et al. "Revisiting Weak-to-Strong Generalization in Theory and Practice: Reverse KL vs. Forward KL." *arXiv preprint arXiv:2502.11107* (2025).

**Experimental Designs Or Analyses:**

The experimental design and results appear to be well-structured.

**Methods And Evaluation Criteria:**

N/A

**Other Comments Or Suggestions:**

I acknowledge the technical challenges involved in using KL divergence in the standard direction, and the use of the opposite direction is somewhat inevitable. However, I hope the authors provide more discussion on this point, perhaps by adding a separate subsection or paragraph.

**Other Strengths And Weaknesses:**

One crucial limitation of the work is the use of KL divergence in the opposite direction. As the authors noted, this deviates from the standard choice. Despite this, the paper's strength lies in the authors' effort to weaken many of the assumptions discussed in Charikar et al. (2024), such as the squared error and the convex strong model class assumption. See the following sections for my suggestions on improving both the strengths and weaknesses.

**Questions For Authors:**

Can the realizability assumption be resolved? For example, is it possible to provide a similar equation using the infimum of the error between strong models and the target model?

Additionally, the main results only show that the misfit is upper-bounded by the gain in weak-to-strong generalization, while the experimental results seem to suggest a positive correlation between the two. Can we also show that the misfit is "lower-bounded" by the gain in weak-to-strong generalization, or is there a case where the misfit is small but the gain in weak-to-strong generalization is large?

If the authors' response successfully addresses my suggestions or questions, I am open to increasing my score.

**Relation To Broader Scientific Literature:**

The paper extends Charikar et al. (2024) by generalizing to Bregman divergence and relaxing the convexity assumption. This contribution enhances the broader understanding of weak-to-strong generalization.

**Theoretical Claims:**

I have checked the proof of the main results, and it appears to be correct.

---

> ### Author Rebuttal · Authors · 2025-04-01
>
> Thank you so much for reading our paper, and for your comments. We address your concerns ahead:
> >...recent references, concurrent works
>
> As the reviewer notes, **the concurrent works listed were released after the ICML submission (and some fairly recently).** Nevertheless, we will be sure to cite these works and include some discussion in our final revision, since they do provide complementary perspectives.
> >...use of KL divergence in the opposite direction...this deviates from the standard choice...more discussion on this point
>
> Thank you for bringing this up, we want to make sure that the ideas can be understood since they are counterintuitive! Our understanding of the differences between forward and reverse KL is based on [this nice blogpost](https://tinyurl.com/2p9zr93x).
>
> Roughly we should think of forward KL as being *mass-seeking*: It prioritizes learning a distribution that covers all possibilities dictated by the teacher. On the other hand, reverse KL is *mode-seeking*: it prioritizes learning a distribution that captures the most frequent behavior in the teacher. One can notice this behavior in how the two loss functions handle a student that disagrees with the teacher and predicts 0% probability for an event: forward KL will be $+\infty$ while reverse KL will exclude that event from the loss. Conversely if the teacher predicts 0% probability for an event and the student disagrees, then forward KL will disregard that event, reverse KL will become $+\infty$.
>
> In the context of weak-to-strong generalization (WTSG), it is plausible that the teacher makes errors. Thus, we *do not* want our student to be mass-seeking. The student should be free to disagree with the teacher. It should be *mode-seeking* as this is likely where most of the signal from the teacher comes from. We will include this discussion (and possibly visualizations) on the reverse KL in the paper.
>
> The fact that the misfit term in the Pythagorean inequality manifests in the reverse direction, shows that the reverse KL is in fact a *better-suited* objective function for WTSG in classification—we show that minimizing reverse KL divergence in the WTS training procedure leads to provable and quantifiable WTSG, whereas minimizing forward KL (i.e., the standard cross-entropy) may not necessarily yield these guarantees. Our result thus shows that there is a certain “directionality” to the correlation between misfit and gain when one goes beyond the squared loss. In fact, we also see (see Rebuttal to 93MC) that the reverse KL setup yields better WTSG experimentally when compared to the standard setup---to this extent, we view the use of reverse KL as a feature rather than a limitation.
> >...realizability assumption
>
> We would like to emphasize that realizability of the ground truth is *not* a critical feature of Theorems 4.1 and 4.3. In particular, note that in these theorems, the weak model has no assumptions placed on it. Thus, we could just as easily take $g$ in Theorems 4.1/4.3 to be the optimal strong model that approximates (with respect to any metric) the possibly non-realizable target function, and we would obtain a misfit-gain inequality with respect to this $g$ (see also Section 5.5, where we do precisely this). This perspective is in line with that taken in Burns et al (2023) where they care about *performance gap recovered (PGR)*, which in some sense, does away with the realizability assumption.
>
> As you mention, a result similar to Theorem 2 in Charikar et al (2024), which incorporates the "distance" of the target function from the strong class into the bound, would be interesting. This needs a little more care in the KL-divergence setting since we don’t have a triangle inequality. We left this result out of our paper to focus more on our Theorem 4.3, which generalizes the misfit-gain inequality to *non-convex* classes of strong models. However, in the reference [3] that you mention, this generalization is worked out.
>
> >...lower-bounding misfit by gain
>
> Lower-bounding the misfit is tricky because we don’t have a triangle inequality for KL divergence. However, in reference [3], such a lower bound was proven, albeit at the cost of a potentially large constant. Without some regularization assumptions, we suspect it would be hard in general to control this constant.

---

> > ### Comment · Reviewer_D6HR · 2025-04-05
> >
> > Thank you for the response. It successfully addresses my concerns. I believe that adding further discussion, particularly regarding the use of reverse KL and connections to concurrent work, would further strengthen the manuscript. Accordingly, I have increased my score to 3 (weak accept).

---

### Official Review · Reviewer_93MC · 2025-03-13

**Overall Recommendation:** 3

**Summary:**

This paper generalizes the conclusion that "performance gain correlates with misfit in weak-to-strong generalization" from prior work on squared loss to Bregman divergence loss. It provides empirical evidence through experiments on synthetic tasks, language tasks, and vision tasks.

**Claims And Evidence:**

The claims made in the submission are supported by clear and convincing evidence.

**Essential References Not Discussed:**

I didn't notice any.

**Experimental Designs Or Analyses:**

The experiments are sound and valid. Notably, compared to prior work (Charikar 2024), they additionally include experiments with language and vision tasks.

**Methods And Evaluation Criteria:**

The datasets and benchmarks are appropriate and make sense for the problem.

**Other Comments Or Suggestions:**

I don't have other comments.

**Other Strengths And Weaknesses:**

Strengths: The theoretical results seem solid and effectively generalize prior work. Additionally, the paper includes experiments on a larger scale and in more commonly relevant domains (language and vision) compared to the initial work on misfit-gain relation.

Weaknesses: My main concerns and questions relate to the practical aspects, which are outlined in the questions section.

**Questions For Authors:**

1. While I appreciate the theoretical contribution of the paper—particularly in how it generalizes the conclusion in Charikar (2024) from squared loss to Bregman divergence, which is a non-trivial task—I still wonder whether it provides any additional insights beyond confirming the correlation between misfit and gain.

2. The authors mention that their theory regarding convex combinations of k strong models provides a "recipe" for minimizing the discrepancy between the weak supervisor and strong student, but “in the opposite direction”. While I understand the mathematical formulation, what is the intuitive interpretation of this being in the opposite direction? How does this impact the results differently compared to simply using the regular loss (i.e., placing the supervisor’s output as the first argument)? Would the standard approach lead to worse results and why?

3. In the experiments (e.g., those in Section 5.2), the authors mention using the reverse KL divergence objective for consistency with the theory while acknowledging that this choice might negatively affect performance. Does this suggest certain impractical aspects of the theoretical framework? Specifically, does the theory require an experimental setup that is not practical in order to produce results that align with its predictions? If the experiments were conducted in a more standard manner, would the same trends and conclusions still hold?

**Relation To Broader Scientific Literature:**

This paper is particularly related to Charikar (2024). It generalizes the prior results on squared loss to Bregman divergence loss, making them applicable to classification tasks. Additionally, compared to Charikar (2024), which conducted synthetic experiments and experiments on molecular predictions, this paper includes experiments involving language and image tasks to validate its results.

**Theoretical Claims:**

I didn’t go through all the proofs in detail, but the theoretical results seem convincing to me.

---

> ### Author Rebuttal · Authors · 2025-04-01
>
> Thank you so much for reading our paper, and for your comments. We are glad that you like our work! We address your concerns ahead:
> >1...additional insights beyond confirming the correlation between misfit and gain.
>
> >2..intuitive interpretation for reverse KL...
>
> We address both 1) and 2) in our response to Reviewer 7oRJ. Please also see our response to Reviewer fGrp for some intuition on the geometry induced by Bregman Divergences.
>
> >3...impractical aspects...If the experiments were conducted in a more standard manner, would the same trends and conclusions still hold?
>
> We thank the reviewer for prompting us to do this comparison; we believe the obtained results lend more conceptual support to our theory. For the experiments in Section 5.2, we ran a comparison where we perform linear-probing for the strong model in the “standard” manner. That is, in the weak-to-strong training phase, instead of finetuning a convex combination of $k=100$ linear layers on the reverse KL objective, we finetune these on the forward cross-entropy objective XE(weak, strong) (i.e., the natural direction), and compare the findings to the numbers presented in Section 5.2. Note that the only difference between the settings now is the objective that the linear layers are finetuned on. The table below shows the comparison for the test accuracy of the strong model, as well the XE loss between the ground truth and the strong model.
> ||cosmos_qa||amazon_polarity||boolq||sciq||CIFAR10||ImageNet||
> |---|---:|---|---:|---|---:|---|---:|---|---:|---|---:|---|
> ||Forward KL|Reverse KL|Forward KL|Reverse KL|Forward KL|Reverse KL|Forward KL|Reverse KL|Forward KL|Reverse KL|Forward KL|Reverse KL|
> |Test Accuracy|0.6378|**0.6407**| 0.8963|**0.8984**|0.6140 |**0.6141**|0.6171|**0.6254**|0.8981|**0.9005**|**0.7133**|0.6986|
> |XE(gt,strong)|0.6147|**0.6141**|0.2788|**0.2570**|**0.6464**|0.6476|0.6439|**0.6436**|0.4052|**0.3361**|**1.4042**|1.4742
>
> The comparison is indeed quite interesting—we can see that the strong model that is finetuned on the **reverse KL objective shows better final test accuracy (i.e., better WTSG) for nearly all the datasets!** Namely, we do not really see significant performance degradation (in fact, we see improvement in nearly all cases) with reverse KL, when compared to the standard setup.
>
> Moreover, we also computed the gain and misfit terms in the Pythagorean inequality, where for the reverse KL experiment, we compute the reverse misfit that we propose (i.e., KL(strong, weak), whereas in the standard forward XE experiment, we compute the the misfit as XE(weak, strong) (i.e., the “natural” misfit that one might consider).
>
> ||cosmos_qa||amazon_polarity||boolq||sciq||CIFAR10||ImageNet||
> |---|---:|---|---:|---|---:|---|---:|---|---:|---|---:|---|
> ||Forward KL|Reverse KL|Forward KL|Reverse KL|Forward KL|Reverse KL|Forward KL|Reverse KL|Forward KL|Reverse KL|Forward KL|Reverse KL|
> |Gain|0.0368|0.0372|0.0717|0.0934|0.0706|0.0694|0.0264|0.0266|0.1772|0.2447|0.5217|0.4517|
> |Misfit|0.6093|0.0613|0.3717|0.1444|0.6319|0.0980|0.6558|0.0961|1.1024|0.3185|3.9392|1.6401|
>
> We can clearly see that if one were to run the standard WTSG setup, the Pythagorean inequality is completely off (gain and misfit don’t quantitatively align), whereas the reverse misfit is more representative of the gain, as confirmed by our theory as well. Again, this indicates a clear “directionality” in the Pythagorean inequality for WTSG in the classification setting! We would be happy to include this discussion in the final version.
>
> It is plausible that in practice, running standard forward XE(weak, strong) minimization might lead to a minimizer that is in fact close to the minimizer of the reverse KL(strong, weak) that we propose. It would be an interesting future direction to precisely characterize scenarios when this is the case.
>
> Finally, we want to emphasize that in our experiments, we only do weak-to-strong finetuning of the last linear layer. In contrast, the best numbers reported in the original WTSG paper by Burns et al. 2023 correspond to full finetuning (on the forward KL objective) of all the weights of the architecture, and the WTS accuracies are hence naturally better. The primary concern we point out about reverse KL minimization is the non-convexity of the objective, which can possibly cause issues for first-order methods. However, if one is finetuning all the weights in the architecture anyway, this concern is not central, since in such a case, even the standard forward cross-entropy objective becomes non-convex.

---

### Official Review · Reviewer_7oRJ · 2025-03-16

**Overall Recommendation:** 2

**Summary:**

This paper generalizes the recent theoretical analysis of weak-to-strong generalization beyond squared loss regression to arbitrary Bregman divergence-based loss functions in the fixed-representation finetuning setting when the strong class is convex.
- For classification tasks, the authors propose to minimize the expected KL divergence between the strong model’s output and the weak labels by optimizing a convex combination of $k$ logistic regression layers on top of the strong model representation. With this special classification objective, it is shown that the gain in weak-to-strong generalization is at least the KL divergence between the strong model and the weak model at the conclusion of weak-to-strong training (i.e. the misfit).
- Empirical results on synthetic, NLP (BoolQ, SciQ, CosmosQA, Amazon Polarity), and vision datasets (CIFAR-10, ImageNet) support the theoretical predictions, demonstrating a correlation between misfit and performance gain.

**Claims And Evidence:**

Yes, claims made in the paper are well-structured, supported by clear theoretical analysis and convincing empirical evidence.

**Essential References Not Discussed:**

To my knowledge, the paper discussed the essential references in the field.

**Experimental Designs Or Analyses:**

I checked the experiments in the main text but not the details in the appendix. The experiments provide sufficient evidence to support the theoretical claims.

**Methods And Evaluation Criteria:**

Yes, the theoretical tools used in the paper are reasonable.

**Other Comments Or Suggestions:**

Some minor questions:
- In the abstract, line 24, the notion of "strong class" is not clear in the context.
- Line 260 (left), what's $\mathcal{F}^*$?

**Other Strengths And Weaknesses:**

Strengths:
- The analysis is well-structured.
- The proposed objective seems interesting (but could be better motivated).
- The empirical evidence is extensive for a theoretical paper.

Weaknesses:
- The intuition behind the key contribution does not seem to be clearly explained in the introduction, lines 071-094. In this paragraph, the authors described a special classification objective whose corresponding misfit is connected to the gain in weak-to-strong generalization. However, the objective is described purely verbally, with neither (a sketching of) the formulation nor any intuitions why the formulation works.
- A major concern that push me toward a negative rating is that the contributions seem marginal.
    1. The main contribution of this work is extending the analysis in Charikar et al. (2024) on the connection between misfit and weak-to-strong generalization gain to the classification setting. The main results in the paper either come with strong assumptions like convexity or are built upon special objectives that could be better motivated.
    2. From a high-level perspective, I appreciate the neat theoretical connection between misfit and weak-to-strong generalization gain as unveiled in Charikar et al. (2024). However, beyond a different objective and the associated theoretical tools, I was having difficulty finding new insights in this work, e.g., why weak-to-strong generalization happens in the classification setting; whether the mechanism is different from the regression setting in Charikar et al. (2024); and how to quantify the gain in weak-to-strong generalization when taking training configurations (like sample complexities, learning dynamics, and architectures) into consideration.

**Questions For Authors:**

Major questions are raised in the "Other Strengths And Weaknesses" section.

**Relation To Broader Scientific Literature:**

The paper builds upon recent theoretical insights by Charikar et al. (2024), which first established a misfit-based characterization of weak-to-strong generalization specifically for regression with squared loss. By employing concepts from information geometry and Bregman divergences, the authors extend this analysis to classification tasks.

**Theoretical Claims:**

I checked all the theoretical statements and proof sketches in the main text, but not all the proofs in the appendix. The main theoretical results in the paper seem reasonable and well-structured.

---

> ### Author Rebuttal · Authors · 2025-04-01
>
> Thank you so much for reading our paper, and for your comments. We address your concerns ahead:
> > ...intuition behind key contribution not clearly explained
>
> Thank you for bringing this up, we want to make sure that the ideas can be understood since they are counterintuitive! Our understanding of the differences between forward and reverse KL is based on [this nice blogpost](https://tinyurl.com/2p9zr93x). We want to note that, **after the ICML deadline**, an [independent work](https://tinyurl.com/mr2jjbr8) also highlighted the value of reverse KL divergence.
>
> Roughly we should think of forward KL as being *mass-seeking*: It prioritizes learning a distribution that covers all possibilities dictated by the teacher. On the other hand, reverse KL is *mode-seeking*: it prioritizes learning a distribution that captures the most frequent behavior in the teacher. One can notice this behavior in how the two loss functions handle a student that disagrees with the teacher and predicts 0% probability for an event: forward KL will be $+\infty$ while reverse KL will exclude that event from the loss. Conversely if the teacher predicts 0% probability for an event and the student disagrees, then forward KL will disregard that event, reverse KL will become $+\infty$.
>
> In the context of weak-to-strong generalization (WTSG), it is plausible that the teacher makes errors. Thus, we *do not* want our student to be mass-seeking. The student should be free to disagree with the teacher. It should be *mode-seeking* as this is likely where most of the signal from the teacher comes from. We will include this discussion (and possibly visualizations) on the reverse KL in the paper.
> >...strong assumptions like convexity...new insights beyond Charikar et al. 2024...sample complexity, dynamics
>
> While the high-level framework of viewing WTSG as “projections onto a convex space” is certainly inspired from the work of Charikar et al. 2024, we want to highlight that our theory in terms of Bregman divergences is more complete, in that it recovers their results for regression as a special case, and also paves the way for other settings. Furthermore, as described in the paper, several things become more complicated while applying this theory to non-symmetric Bregman divergences. We believe that our technical contributions towards fixing these yield insights that are of interest by themselves.
>
> First, the fact that the misfit term in the Pythagorean inequality manifests in the reverse direction, shows that the reverse KL is a *better-suited* objective function for WTSG in classification—we show that minimizing reverse KL divergence in the WTS training procedure leads to provable and quantifiable WTSG, whereas minimizing forward KL (i.e., the standard cross-entropy) may not necessarily yield these guarantees. In fact, we also see (see Rebuttal to 93MC) that the reverse KL setup yields better WTSG experimentally when compared to the standard setup. Our result thus shows that there is a certain “directionality” to the correlation between misfit and gain when one goes beyond the squared loss.
>
> Second, our guarantee (Theorem 4.3) for reverse KL minimization, **holds even if the function space is non-convex** (unlike the work of Charikar et al. 2024). Our solution to handle non-convexity, namely optimizing over a convex combination of $k$ functions so as to approximate the convex hull of the function space—is both practical (as evidenced by the experimental results), and also allows us to derive concrete theoretical guarantees. The error term in the guarantee becomes smaller as $k$ increases, matching the intuition that the convex hull is being approximated better. **Importantly, this solution allows us to optimize over convex combinations of functions from an arbitrary (possibly non-convex) function space, and not just linear/logistic heads!** This opens the gates for obtaining WTSG guarantees with *full-finetuning*, and not just linear probing. In particular, the same bound holds even if we finetune all the weights of a convex combination of neural networks—such freedom is not afforded in the theory of Charikar et al. 2024. Our solution requires deriving an insightful connection with existing density estimation methods, and requires non-trivial adaptations to our setting. We view this contribution to be novel, and possibly also as a useful modeling paradigm in other learning settings, especially given that our experimental numbers improve with reasonably large values of $k$.
>
> Finally, sample complexity bounds are a straightforward adaptation from Charikar et al. (2024), and we wanted to keep the paper focused on new contributions. As for learning dynamics, for our initial study, we assume that the $\arg\min$ of the reverse KL can be approximated well (as also suggested by our experiments); nevertheless, this is an interesting future direction.
>
> >...what's $\mathcal{F}*$
>
> It is the dual space of $\mathcal{F}$ (see Line 123).

---

> > ### Comment · Reviewer_7oRJ · 2025-04-03
> >
> > Thanks for the responses. I partially agree with the authors that, from a technical perspective, extensions beyond convex functions and linear probing are valuable in some sense. However, subjectively, I don't find these extensions alone to be strong enough for an ICML paper. I also agree that the new perspective on forward and reverse KL could be insightful. But I don't see how this new perspective can be naturally integrated into the original story based on the special objective proposed. Overall, I believe the paper would benefit from a major revision of its main messages and storyline. Therefore, I will maintain my current score.

---

### Decision · Program_Chairs · 2025-05-01

**Decision:**

Accept (poster)

**Comment:**

The paper addresses a very interesting problem using an original tool for the analysis (Bregman divergences). The reviews make the paper stand borderline so I read the paper.

Reviewer 7oRJ asks for a motivation of the tool used, which is indeed a fair question for a technical paper employing a "tangential" approach to the SOTA. The authors did not do an especially good job at answering this question, choosing to justify it by a property of the functions (the projection property) making the result (Cor 4.2) look nice in context, rather than an axiomatic reply: "why indeed should we use Bregman divergences generally ?". Fortunately, the seminal paper by Leonard J. Savage "Elicitation of personal probabilities end expectations" answers the reviewer's question: Bregman divergences have a specific analytical form desirable for properness. Another justification follows from the paper "Clustering with Bregman divergences" by Banerjee et al.: Bregman divergences fully axiomatize the best constant parameter approximation to be the population average, which makes sense in the context. An the work of Reid and Williamson (2 JMLR papers) will definitely conclude this "why" question in support of the authors' approach.

Being familiar with Bregman divergences I have particularly appreciated Th 4.1, which gives consistency to the "switching arguments" property as a core property of (some) Bregman divergences.

I also appreciated the authors' honesty in pointing to concurrent work. I have checked it and at least from a theoretical standpoint, the authors' work is superior.

Therefore, I would tend to accept the paper under the condition that the authors should definitely polish their narrative to support their approach -- it is definitely suboptimal as it stands (see above).